# ADAPTIVE AUGMENTATION-AWARE LATENT LEARNING FOR ROBUST LiDAR SEMANTIC SEGMENTATION

**Wangkai Li[1], Zhaoyang Li[1], Yuwen Pan[1], Rui Sun[1], Yujia Chen[1], Tianzhu Zhang[1,2]***

[1]University of Science and Technology of China
[2]National Key Laboratory of Deep Space Exploration, Deep Space Exploration Laboratory
{`lwklwk, lizhaoyang, panyw, issunrui, yujia_chen`}@mail.ustc.edu.cn,
 `tzzhang@ustc.edu.cn`

## ABSTRACT

Adverse weather conditions significantly degrade the performance of LiDAR point cloud semantic segmentation networks by introducing large distribution shifts. Existing augmentation-based methods attempt to enhance robustness by simulating weather interference during training. However, they struggle to fully exploit the potential of augmentations due to the trade-off between minor and aggressive augmentations. To address this, we propose A3Point, an adaptive augmentation-aware latent learning framework that effectively utilizes a diverse range of augmentations while mitigating the semantic shift, which refers to the change in the semantic meaning caused by augmentations. A3Point consists of two key components: semantic confusion prior (SCP) latent learning, which captures the model's inherent semantic confusion information, and semantic shift region (SSR) localization, which decouples semantic confusion and semantic shift, enabling adaptive optimization strategies for different disturbance levels. Extensive experiments on multiple standard generalized LiDAR segmentation benchmarks under adverse weather demonstrate the effectiveness of our method, setting new state-of-the-art results.

## 1 INTRODUCTION

LiDAR semantic segmentation is vital for 3D vision tasks such as autonomous driving (Li & Ibanez-Guzman, 2020; Li et al., 2020; Aksoy et al., 2020; Zhao et al., 2023). However, existing methods (Ando et al., 2023; Choy et al., 2019; Lai et al., 2023; Puy et al., 2023) often struggle in adverse weather (e.g., fog, snow, and rain) due to severe distribution shifts in point clouds, causing a mismatch between training and testing data. Since most outdoor scene point cloud datasets (Behley et al., 2019; Fong et al., 2022; Xiao et al., 2022b) are collected in normal weather, developing a robust network that generalizes across diverse conditions is increasingly crucial. Addressing this challenge is essential for achieving reliable, weather-invariant LiDAR semantic segmentation.

To mitigate performance degradation, existing studies (Xiao et al., 2023; Park et al., 2024; Kong et al., 2023b) enhance network robustness by simulating adverse weather during training via simulation-based or augmentation-based approaches. Simulation-based methods (Bijelic et al., 2018; Hahner et al., 2022; 2021) model physical equations to replicate weather effects on point clouds but require separate modeling for each condition, making it impractical to cover all variations. Augmentation-based methods (Xiao et al., 2023; Park et al., 2024; Kim et al., 2023) introduce geometric perturbations and point drop to mimic weather-induced distortions more flexibly. However, they remain underutilized for two reasons (Fig. 1): (1) mild augmentations fail to generalize to severe conditions, while (2) excessive augmentations distort point cloud distribution, causing semantic shift (Wang et al., 2021; Bai et al., 2022; Yuan et al., 2021), where augmented regions no longer align with original semantics, thereby hindering training. This dilemma leads existing methods to restrict augmentation range and magnitude, limiting their potential. *How to utilize a larger augmentation space while mitigating semantic shift remains a compelling challenge.*

---

*Corresponding author

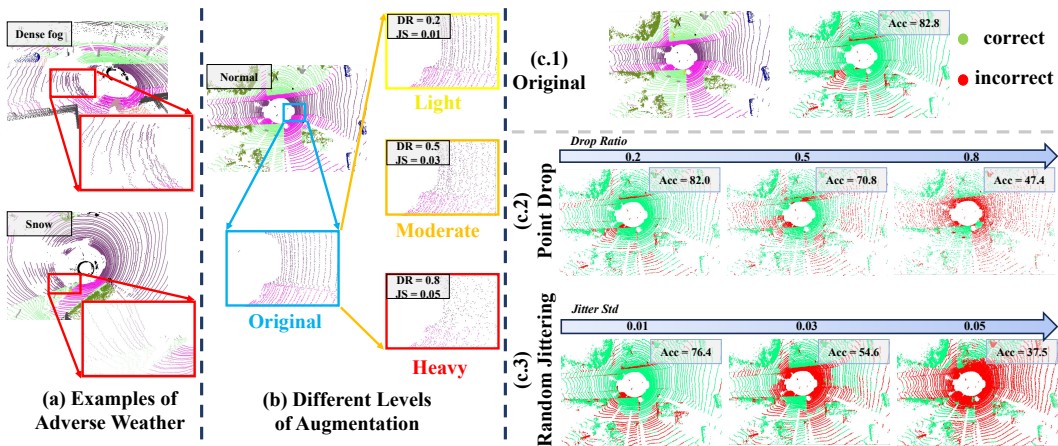

Figure 1: (a) Point cloud distortions caused by adverse weather conditions. (b) Augmentation at different levels (light, moderate, and heavy), where we adjust the drop ratio (DR) for point drop and jitter std (JS) for random jittering. (c) Visualization of segmentation accuracy under different augmentation levels, showing that aggressive distortions lead to significant performance degradation.

This analysis motivates us to explore a broader, more aggressive augmentation space to better simulate weather-induced distortions at varying intensities. However, ensuring its effectiveness requires addressing the potential semantic shift in augmented point clouds. Directly modeling semantic shift is difficult, as prediction errors arise from two factors: (1) **semantic confusion** (Fig.2 (a)), which is the network's inherent property that struggles to distinguish similar classes (e.g., *road* vs. *sidewalk*) despite correct labels; (2) **semantic shift** (Fig.2 (b)), where excessive augmentation distorts point cloud distributions and the original labels fail to describe the corresponding regions. For semantic confusion, original labels should be preserved to enhance the network's discriminative ability. In contrast, semantic shift requires adapting supervision to avoid misleading signals. Thus, the key to solving semantic shift lies in disentangling these two factors in augmented point clouds.

In this paper, we model semantic shift in augmented point clouds to enable their effective use during training. Observing that semantic confusion exists in both raw and augmented data and is consistent across domains, while semantic shift occurs only in augmented data, we propose a two-step strategy to identify it: (1) Mining semantic confusion priors from normal-condition predictions. Inspired by VQVAE (Van Den Oord et al., 2017), we frame this as a **discrete latent representation learning task**: class-specific local confusion patterns are encoded into a latent space and represented by quantized latent variables, with representational capacity enforced via reconstruction. (2) Detecting semantic shift as anomalies in augmented point clouds. We apply the learned latent encoder to augmented predictions, formulating semantic shift localization as an **anomaly detection problem**: by comparing augmented latent representations with the learned priors, we distinguish semantic consistency regions (affected only by semantic confusion) from semantic shift regions (additionally affected by semantic shift). This separation enables targeted optimization during training.

Based on the above discussion, we propose **A**daptive **A**ugmentation-**A**ware latent learning for robust LiDAR semantic segmentation in adverse conditions, namely A3Point, which involves two key components: semantic confusion prior (SCP) latent learning and semantic shift region (SSR) localization, to fully explore the potential of a large and diverse augmentation space for robust point cloud segmentation. **To capture the network's inherent semantic confusion**, in the SCP latent learning module, we perform class-wise latent encoding on predictions from original point clouds, using quantized latent variables to represent local confusion patterns. Through vector quantization and reconstruction constraints, this process learns meaningful, representative embeddings capable of reconstructing prediction maps. **To disentangle semantic confusion from semantic shift**, in the SSR localization module, we dynamically track the representation distribution of each quantized latent variable and use a frozen prior latent encoder to implicitly represent augmented predictions. By treating semantic shift detection as an anomaly detection problem, we adaptively distinguish semantic consistency regions (SCR) from semantic shift regions (SSR). In SCR, original labels remain effective for optimization, while in SSR, we apply knowledge distillation, selecting the global nearest quantized latent variable as a supervisory constraint. By jointly localizing SSR and adapt-

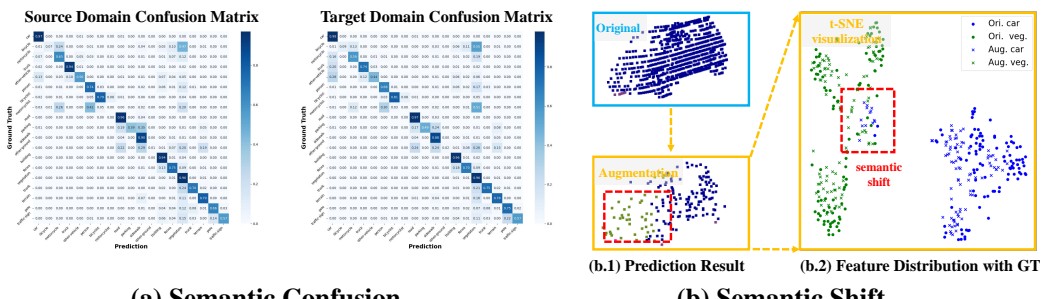

Figure 2: Demonstration of **semantic confusion** and **semantic shift**. (a) Confusion matrices from source (normal weather) and target (adverse weather) domains. Despite domain shifts, semantic confusion remains consistent. (b) Aggressive augmentations alter point cloud density and shape, leading to semantic misalignment (e.g., *car → veg.*).

ing optimization strategies accordingly, A3Point fully harnesses diverse augmentations to improve segmentation robustness under varying disturbance levels.

Our contributions are as follows: 1) We introduce a novel perspective to overcome limitations of augmentation-based approaches, enabling the effective utilization of a large and diverse augmentation space to improve LiDAR segmentation robustness under adverse weather. 2) We propose a two-step framework to decouple semantic confusion and mitigate semantic shift, comprising semantic confusion prior (SCP) latent learning and semantic shift region (SSR) localization. 3) We validate our approach through extensive experiments on multiple standard generalized LiDAR segmentation benchmarks under adverse weather, achieving new state-of-the-art results.

## 2 RELATED WORK

### 2.1 3D POINT CLOUD SEMANTIC SEGMENTATION

Semantic segmentation of 3D point clouds assigns a label to each point and is typically approached via three paradigms: point-based, projection-based, and voxel-based. Point-based methods directly take 3D points as input. PointNet (Qi et al., 2017) was the first to introduce this approach, utilizing multi-layer perceptrons to extract per-point features. Subsequent works (Thomas et al., 2019; Zhao et al., 2021; Choe et al., 2022; Fan et al., 2021; Li et al., 2024; 2025e) further advanced this paradigm. While these methods minimize information loss and achieve strong performance, they demand high computational resources when applied to large-scale LiDAR data. Projection-based methods (Ando et al., 2023; Kong et al., 2023a; Milioto et al., 2019; Xiao et al., 2021; Zhang et al., 2020) map point clouds to 2D range-view images via spherical projection, enabling the use of 2D segmentation networks. These methods are computationally efficient but suffer from information loss during projection, leading to slightly lower accuracy. Voxel-based methods (Choy et al., 2019; Lai et al., 2023; Zhu et al., 2021; Graham et al., 2018; Tang et al., 2020) divide 3D point clouds into sparse voxel grids and aggregate points within the same voxel. The introduction of sparse convolutions significantly reduces computational costs, making this paradigm well-suited for large-scale outdoor LiDAR scenes. In this work, we adopt voxel-based architectures (Choy et al., 2019; Tang et al., 2020) as our baseline to balance inference efficiency and segmentation performance.

### 2.2 LIDAR UNDER ADVERSE WEATHER CONDITIONS

In real-world applications, robust scene understanding under adverse weather is critical, especially given the safety demands of autonomous driving (Kong et al., 2023b; Xiao et al., 2023; Sakaridis et al., 2021; Pan et al., 2025). However, extreme weather can significantly disturb point cloud distributions (Filgueira et al., 2017; Heinzler et al., 2019; Peynot et al., 2009; Ryde & Hillier, 2009), leading to a dramatic drop in the performance of LiDAR segmentation networks. To mitigate domain discrepancy between normal and adverse weather conditions, unsupervised domain adaptation (UDA) approaches (Hahner et al., 2022; 2021; Luo et al., 2021; Xiao et al., 2022a;b; Yang et al., 2021) have been explored. Drawing inspiration from semi-supervised learning (Sun et al.,

2025a;c;b), these methods leverage labeled source domain data and unlabeled target domain data to learn domain-invariant features, improving cross-domain performance (Li et al., 2025b;c;a;d; 2026; Chen et al., 2025b;a; He et al., 2025a;b). However, UDA-based methods are limited to specific and visible target domains, making them insufficient for generalizing to unknown weather disturbances. In this paper, we adopt the domain generalization setting (Kim et al., 2024; 2023; Li et al., 2023a;b), aiming to train a model using a single source domain under normal weather conditions, without access to target data from adverse weather during training.

## 2.3 AUGMENTATION FOR ROBUST LiDAR SEGMENTATION

To enhance the robustness of LiDAR segmentation models, existing methods introduce point cloud corruptions during training to simulate the interference caused by adverse weather. Simulation-based methods (Bijelic et al., 2018; Hahner et al., 2022; 2021) rely on prior weather knowledge to construct physical models that artificially simulate point clouds under adverse conditions. Rather than explicitly modeling specific weather effects, augmentation-based methods (Xiao et al., 2023; Park et al., 2024; Kim et al., 2023) provide a more general and flexible approach. Inspired by 2D image augmentation, Mix-based methods (Kong et al., 2023c; Nekrasov et al., 2021; Xiao et al., 2022a; Zhao et al., 2024) blend two LiDAR scans to enhance training diversity. To better simulate disturbances caused by adverse weather, recent works (Park et al., 2024; Xiao et al., 2023) identify two primary degradation patterns: (1) geometric perturbation and (2) point drop. PointDR (Xiao et al., 2023) primarily introduces the SemanticSTF benchmark and an accompanying augmentation-based baseline that simulates weather-induced disturbances via geometric perturbation and point dropping. LiDARWeather (Park et al., 2024) proposes a learnable Point Drop strategy for adaptive augmentation. However, these methods remain constrained by a limited perturbation space. In contrast, our approach explores a broader range of perturbations and explicitly addresses the semantic shift problem caused by aggressive augmentation, enabling more robust network training.

## 3 METHOD

### 3.1 PROBLEM DEFINITION

In domain generalization (DG) for LiDAR semantic segmentation, the network is trained on labeled source domain data and need to be generalized to unseen target domain data. To be specific, the source domain can be denoted as $D_s = \{(x_i^S, y_i^S)\}_{i=1}^{N_S}$, where $x_i^S \in X_S$ represents a LiDAR point cloud scan with $y_i^S \in Y_S$ as the corresponding point-wise one-hot label covering $C$ classes. The target domain can be denoted as $D_t = \{(x_i^T)\}_{i=1}^{N_T}$, where target label $Y_T$ shares the same label space with $Y_S$. Since the target domain is not accessible during the training process, we omit the superscript $S/T$ in the following notation for brevity.

### 3.2 PRELIMINARIES

**Augmentation-based Training for DG.** Existing methods explore weather-induced disturbances and apply them as data augmentation. This approach can be viewed as a domain randomization paradigm for learning a domain-generalizable network. During training, the loss is first computed on the original scan to train a neural network $f$:

$$\mathcal{L}_{ce} = \frac{1}{N_S} \sum_{i=1}^{N_S} \frac{1}{n_i} \sum_{j=1}^{n_i} \ell_{ce}(f(x_{ij}), y_{ij}), \tag{1}$$

where $\ell_{ce}$ denotes the voxel-wise cross-entropy loss, and $n_i$ is the number of valid voxels in $x_i$. Then, for each $x_i$, random augmentations are simultaneously applied to $(x_i, y_i)$, obtaining $(\hat{x}_i, \hat{y}_i) = \mathcal{A}\{(x_i, y_i)\}$. The augmented training pair is also implemented through the cross-entropy loss:

$$\hat{\mathcal{L}}_{ce} = \frac{1}{N_S} \sum_{i=1}^{N_S} \frac{1}{\hat{n}_i} \sum_{j=1}^{\hat{n}i} \ell_{ce}(f(\hat{x}_{ij}), \hat{y}_{ij}). \tag{2}$$

The total training loss can be represented as: $\mathcal{L} = \mathcal{L}_{ce} + \hat{\mathcal{L}}_{ce}$.

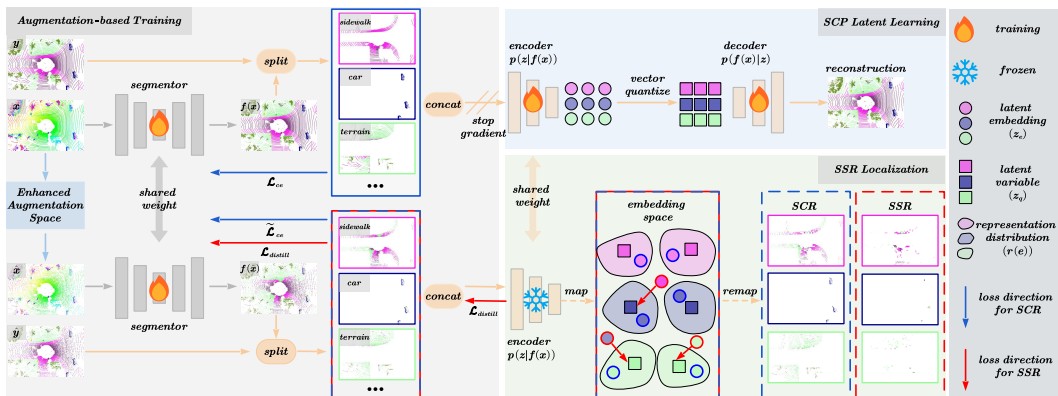

Figure 3: Pipeline of A3Point. We explore an abundant augmentation space (Sec.3.4) and propose two key components: SCP latent learning to capture inherent semantic confusion (Sec.3.5) and SSR localization to decouple semantic shift (Sec.3.6).

**Vector Quantized Variational AutoEncoder.** VQ-VAE (Van Den Oord et al., 2017) is a variant of variational autoencoder (Kingma et al., 2013) that learns a discrete latent representation. It consists of an encoder $\mathbb{E}$, a decoder $\mathbb{D}$, and a codebook $\mathcal{C} = \{e_1, e_2, ..., e_K\}$ containing $K$ learnable embeddings. The encoder maps the input $x$ to a continuous latent representation $z_e = \mathbb{E}(x)$, which is then quantized to the nearest codebook entry $e_k$ using a nearest-neighbor lookup (we use a single random variable $z$ to represent the discrete latent variables for simplicity):

$$z_q = \text{quantize}(z_e) = e_k, \text{where } k = \arg\min_j ||z_e - e_j||_2. \tag{3}$$

The decoder reconstructs the input from the quantized latent representation: $\bar{x} = \mathbb{D}(z_q)$. Total training objective is:

$$\mathcal{L} = ||x - \bar{x}||_2^2 + ||\text{sg}(z_e) - z_q||_2^2 + ||z_e - \text{sg}(z_q)||_2^2, \tag{4}$$

which consists of reconstruction loss, codebook loss, and commitment loss, where $\text{sg}(\cdot)$ denotes the stop-gradient operation. The codebook loss brings the selected latent variables $e$ close to encoder outputs, while the commitment loss encourages the encoder to produce latent representations close to the codebook entries. The discrete latent representation learned by VQ-VAE can capture the underlying structure of the data and has been successfully applied in various tasks, such as image generation (Razavi et al., 2019; De Fauw et al., 2019; Yu et al., 2021) and unsupervised representation learning (Liu et al., 2023; Chen et al., 2024; Takida et al., 2023).

### 3.3 OVERVIEW OF A3POINT FRAMEWORK

We first define the enhanced augmentation space used during training (Sec.3.4). Then, we model a discrete latent representation learning process to learn semantic confusion prior (Sec.3.5). Next, we localize the semantic shift regions through a form of anomaly detection (Sec.3.6). Finally, we introduce region-specific optimization strategies (Sec.3.7). Our overall framework is shown in Fig.3.

### 3.4 ENHANCED AUGMENTATION SPACE

Previous works (Xiao et al., 2023; Park et al., 2024) identify that the main disturbances caused by adverse weather can be summarized as (1) geometric perturbation, caused by perceived distance shifts, and (2) point drop, resulting from beam attenuation, beam missing, or potential occlusions. This motivates us to use random jittering and point drop as the primary and generic augmentation strategies. Unlike previous methods that only adopt limited range and magnitude, we define a broader augmentation space to fully simulate various levels of weather disturbances. Specifically, we define jitter std in a range of $[j_{min}, j_{max}]$ for random jittering and drop ratio range of $[d_{min}, d_{max}]$ for point drop, and uniformly sample the perturbation magnitudes during training. For other subsidiary augmentations, we follow previous works and employ random rotation, random scaling, random flipping, random noise perturbation, scan mix (Kong et al., 2023c; Xiao et al., 2022a).

### 3.5 SEMANTIC CONFUSION PRIOR LATENT LEARNING

Semantic confusion, which refers to the network's inherent uncertainty in distinguishing between classes, as reflected in the predicted probability distribution. To decouple semantic confusion and semantic shift in the augmented point cloud, we mine prior knowledge of semantic confusion from the original point cloud predictions.

Specifically, we first introduce an autoencoder for prior latent learning, which follows VQ-VAE (Van Den Oord et al., 2017). Through quantized latent variables and a reconstruction process, it can effectively learn meaningful representations with sufficient representational power. The input for encoder $\mathbb{E}$ is obtained by first concatenating $f(x)$ (processed through softmax) and $x$, then splitting the result class-wise according to the label $y$ to prevent inter-class interactions, and finally concatenating the resulting submatrices along the batch dimension. This can be expressed as:

$$z_e = \mathbb{E}(\underbrace{[f(x^1) \oplus x^1, f(x^2) \oplus x^2, ..., f(x^C) \oplus x^C]}_{n \times (C+3)}),\tag{5}$$

where $f(x^i) \oplus x^i$ represents the concatenation of prediction and coordinates for points with class $i$, and $n$ is the number of valid voxels. The $[\cdot, \cdot]$ notation denotes the concatenation operation along the batch dimension. The output for decoder $\mathbb{D}$ is reconstructed to $[f(x^1); f(x^2); ...; f(x^C)]$. Eq.4 is used to optimize for this process without backpropagating the gradients to $f$.

Compared to the $K \times D$ codebook in VQ-VAE, we use class-specific sub-codebooks of size $C \times k \times D$, where $k$ is the number of latent variables in each sub-codebook, and $D$ is the dimension of the latent variables. Each latent variable in the sub-codebook can be considered as modeling a specific local semantic distribution pattern under corresponding class. The encoder $\mathbb{E}$ models the $p(z|f(x))$, which represents the prior distribution of the latent variables $z$ given the predicted probabilities $f(x)$. The decoder $\mathbb{D}$ models the $p(f(x)|z)$, which represents the posterior distribution of reconstructing the predicted probabilities $f(x)$ conditioned on the latent variables $z$. Through this process, the autoencoder can model the network's semantic confusion online during the training process.

### 3.6 SEMANTIC SHIFT REGION LOCALIZATION

Excessive augmentation can cause semantic shift in certain regions, leading to abnormal prediction results, as shown in Fig. 2 (b). After modeling the semantic confusion prior, we can assess whether the network's predictions for the augmented point cloud conform to the normal semantic distribution patterns, thus treating the localization of semantic shift regions as an anomaly detection problem.

During latent learning process, we dynamically track the representation distribution of each quantized latent variable. For each $e_i$, we maintain statistics of its corresponding latent embedding distribution before quantization and store variance $\sigma_i^2$ of each channel, which is updated using exponential moving average. The representation distribution of each $e_i$ can be expressed as:

$$r(e_i) = \mathcal{N}(e_i, \mathrm{diag}(\sigma_{i,1}^2, \sigma_{i,2}^2, \ldots, \sigma_{i,D}^2)),\tag{6}$$

Next, we use the frozen prior latent encoder $\mathbb{E}$ to map predictions of augmented point cloud to latent embedding space. Since the encoder models the class-wise $p(z|f(x))$, embeddings from regions affected by semantic shift will not fall into the representation distribution of their corresponding sub-codebooks. We locate the semantic shift regions by checking whether the embeddings lie within the representation distribution of their nearest latent variable in corresponding sub-codebooks.

After distinguishing all the embeddings, we remap them back to the prediction space, i.e., $f(\hat{x})$, to determine the semantic consistency regions (SCR) and semantic shift regions (SSR) in the prediction space. We use two masks to represent these two regions:

$$M_{SCR} = \mathbb{1}(z_e \in r(NN_{sub}(z_e))),\tag{7}$$

$$M_{SSR} = \mathbb{1}(z_e \notin r(NN_{sub}(z_e))),\tag{8}$$

where $z_e$ represents the latent embeddings of the augmented point cloud predictions, $NN_{sub}(z_e)$ denotes the nearest latent variable of $z_e$, and $\mathbb{1}(\cdot)$ is the indicator function.

### 3.7 OPTIMIZATION STRATEGIES

After determining the SCR and SSR, we assign different optimization strategies. For SCR, we use the original labels:

$$\widetilde{\mathcal{L}}_{ce} = \ell_{ce}(f(\hat{x}) \odot M_{SCR}, y \odot M_{SCR}) = \hat{\mathcal{L}}_{ce} \odot M_{SCR}, \tag{9}$$

where $\odot$ denotes element-wise multiplication. For SSR, we propose a latent variable-based distillation loss to provide appropriate supervisory signals. Instead of querying the nearest neighbor from sub-codebook of the corresponding class, we obtain the closest semantic confusion pattern prior for this region by querying from global codebook. Then, we use this prior to distill the latent embeddings of SSR:

$$\mathcal{L}_{distill} = ||z_e - sg[NN_{global}(z_e)]||_2^2, \tag{10}$$

where $NN_{global}(z_e)$ denotes the global nearest neighbor latent variable. Eq.10 is similar to previous commitment loss while we do not update the encoder $\mathbb{E}$, but backpropagate the gradient to the network $f$. Our total training loss is:

$$\mathcal{L}_{total} = \mathcal{L}_{ce} + \widetilde{\mathcal{L}}_{ce} + \lambda \mathcal{L}_{distill}, \tag{11}$$

where $\lambda$ is a hyperparameter balancing loss terms. In this way, the semantic confusion prior learned from the original point cloud can provide meaningful supervisory signals for semantic shift regions, enhancing the model's performance on augmented data while preserving semantic consistency. For further implementation details, discussions and algorithm flow, please refer to Appendix B-E.

## 4 EXPERIMENTS

### 4.1 EXPERIMENTAL SETUP

**Datasets.** We use four datasets: SemanticKITTI (Behley et al., 2019), SynLiDAR (Xiao et al., 2022b), SemanticKITTI-C (Kong et al., 2023b), and SemanticSTF (Xiao et al., 2023). ❶ SemanticKITTI: 19,130 training scans (sequences 00-10, except 08), collected in urban environments under standard weather conditions. ❷ SynLiDAR: 198,396 synthetic scans (19 billion points) generated using Unreal Engine 4. ❸ SemanticKITTI-C: Corrupted version of SemanticKITTI generated via simulation. ❹ SemanticSTF: 2,076 LiDAR scans from STF (Bijelic et al., 2020) under adverse weather (snow, dense fog, light fog, rain), split into 1,326 training, 250 validation, and 500 testing scans. SemanticKITTI and SynLiDAR serve as source domains, while SemanticKITTI-C and SemanticSTF assess robustness as target domains. We denote SemanticKITTI as [A], SynLiDAR as [B], SemanticSTF as [C], and SemanticKITTI-C as [D] for brevity.

**Evaluation Metrics.** We adopt MinkowskiNet-18/32width (Choy et al., 2019) as the base model and also evaluate SPVCNN (Tang et al., 2020). Performance is measured by Intersection over Union (IoU) per class and mean IoU (mIoU) across classes, including breakdowns by weather conditions.

**Implementation Details.** The segmentation network is trained with SGD (learning rate 0.24, weight decay 0.0001), and the autoencoder with Adam (learning rate 0.001). The sub-codebook size $k$ is set to 32. These networks are updated alternately. For augmentation, we set the drop ratio to [0.2, 0.8] and jitter standard deviation to [0.01, 0.05]. Training runs for 50 epochs with a batch size of 4 on an RTX 3090 (24 GB). The balancing coefficient $\lambda$ is set to 0.1 to maintain gradient stability.

### 4.2 COMPARISON WITH EXISTING METHODS

**Overall Quantiative Results.** Tab. 1-2 compare A3Point with state-of-the-art domain generalization methods on the $[A] \rightarrow [C]$ and $[B] \rightarrow [C]$ benchmarks. The baseline is trained only with $\mathcal{L}_{ce}$. A3Point outperforms all methods, achieving 9.9% and 11.7% mIoU improvements over the baseline. Class-wise analysis shows that A3Point performs particularly well on safety-critical classes such as *motorcycle*, *bicyclist*, *traffic sign*, and *car*, which are typically challenging due to unique geometries and susceptibility to adverse conditions. The local pattern encoding in VQ-VAE and region-specific optimization enhance robust feature capture for these classes under diverse weather conditions.

**Weather-level Comparison.** As shown in Tab. 1-2, A3Point demonstrates superior robustness across all weather conditions, with substantial leads even in the most severe cases like dense fog

Table 1: Comparison results of [A] → [C]. ∗ denotes the reproduced result with the same backbone.

| Methods | car | bi.cle | mt.cle | truck | oth-v. | pers. | bi.clst | mt.clst | road | parki. | sidew. | other-g. | build. | fence | veget. | trunk | terr. | pole | traf. | D-fog | L-fog | Rain | Snow | mIoU | gain |
|---|---|---|---|---|---|---|---|---|---|---|---|---|---|---|---|---|---|---|---|---|---|---|---|---|---|
| Oracle | 89.4 | 42.1 | 0.0 | 59.9 | 61.2 | 69.6 | 39.0 | 0.0 | 82.2 | 21.5 | 58.2 | 45.6 | 86.1 | 63.6 | 80.2 | 52.0 | 77.6 | 50.1 | 61.7 | 51.9 | 54.6 | 57.9 | 53.7 | 54.7 | - |
| Baseline | 67.1 | **5.0** | 28.1 | 38.5 | 14.6 | 45.8 | 8.3 | 13.8 | 40.1 | 16.1 | 26.1 | 3.3 | 71.6 | 52.7 | 53.8 | 33.9 | 39.2 | 25.3 | 12.7 | 30.7 | 30.1 | 29.7 | 25.3 | 31.4 | +0.0 |
| PointDR∗ (Xiao et al., 2023) | 69.2 | 1.0 | 8.9 | **41.9** | 7.6 | 48.9 | 17.0 | 36.2 | 57.8 | 15.9 | 32.3 | 4.0 | 75.7 | 46.4 | 54.0 | 36.2 | 43.9 | 23.7 | 24.2 | 37.3 | 33.5 | 35.5 | 26.9 | 33.9 | +2.5 |
| DGUIL (Kim et al., 2023) | 78.2 | 2.5 | 33.0 | 29.7 | 6.1 | 49.8 | 0.8 | 40.9 | 67.3 | 7.2 | 38.0 | 2.2 | **79.8** | **54.4** | **64.1** | 36.8 | 52.3 | 31.0 | 40.0 | 36.3 | 34.5 | 35.5 | 33.3 | 37.6 | +6.2 |
| WADG (Du et al., 2024) | 72.0 | 0.0 | 32.9 | 37.0 | 1.9 | 37.7 | 6.8 | 52.9 | 59.9 | 10.7 | 31.8 | 2.2 | 76.0 | 48.8 | 62.7 | 34.0 | 49.3 | 23.6 | 20.4 | 39.5 | 32.5 | 31.7 | 29.4 | 34.8 | +3.4 |
| DGLSS (He et al., 2024) | 69.6 | 0.8 | 42.8 | 34.4 | 8.9 | 41.9 | 12.8 | 44.5 | 52.0 | 14.5 | 30.8 | **6.0** | 77.8 | 51.1 | 57.6 | 38.9 | 43.2 | 29.7 | 30.6 | 34.2 | 34.8 | 36.2 | 32.1 | 36.2 | +4.8 |
| LiDARWeather (Park et al., 2024) | 86.1 | 4.8 | 13.8 | 39.7 | **26.6** | **55.4** | 8.5 | 50.4 | 63.7 | 14.9 | 37.9 | 5.5 | 75.2 | 52.7 | 60.4 | 39.7 | 44.9 | 30.1 | 40.8 | 36.0 | 37.5 | 37.6 | 33.1 | 39.5 | +8.1 |
| NTN (Park et al., 2025) | 83.3 | 3.7 | 31.3 | 36.2 | 18.2 | 53.3 | 6.8 | **55.9** | 67.2 | **18.1** | 37.2 | 5.4 | 72.1 | 41.8 | 58.0 | 36.0 | 46.0 | 28.2 | 39.8 | 35.3 | 35.1 | 35.7 | 32.4 | 38.9 | +7.5 |
| A3Point (ours) | **88.3** | 4.1 | **57.5** | 29.0 | 7.6 | 45.3 | **24.0** | 46.4 | **69.2** | 16.9 | **38.4** | 3.3 | 74.8 | 48.2 | 63.1 | **42.9** | **49.2** | **32.6** | **41.8** | **41.1** | **38.5** | **38.2** | **37.2** | **41.3** | +9.9 |

Table 2: Comparison results of [B] → [C]. ∗ denotes the reproduced result with the same backbone.

| Methods | car | bi.cle | mt.cle | truck | oth-v. | pers. | bi.clst | mt.clst | road | parki. | sidew. | other-g. | build. | fence | veget. | trunk | terr. | pole | traf. | D-fog | L-fog | Rain | Snow | mIoU | gain |
|---|---|---|---|---|---|---|---|---|---|---|---|---|---|---|---|---|---|---|---|---|---|---|---|---|---|
| Oracle | 89.4 | 42.1 | 0.0 | 59.9 | 61.2 | 69.6 | 39.0 | 0.0 | 82.2 | 21.5 | 58.2 | 45.6 | 86.1 | 63.6 | 80.2 | 52.0 | 77.6 | 50.1 | 61.7 | 51.9 | 54.6 | 57.9 | 53.7 | 54.7 | - |
| Baseline | 33.8 | 1.7 | 3.3 | 15.5 | 0.2 | 25.5 | 1.6 | 3.4 | 15.3 | 9.2 | 16.8 | 0.1 | 33.4 | 21.9 | 39.5 | 18.7 | **44.0** | 8.8 | 0.8 | 15.2 | 16.0 | 16.8 | 12.8 | 15.5 | +0.0 |
| PointDR∗ (Xiao et al., 2023) | 41.1 | 2.8 | 3.4 | 18.1 | 0.2 | 31.3 | **2.8** | 3.3 | 34.4 | **10.2** | 19.7 | 1.0 | 52.7 | 22.0 | 48.5 | 21.3 | 38.3 | 19.2 | 5.6 | 19.1 | 20.3 | 25.3 | 19.0 | 19.8 | +4.3 |
| WADG (Du et al., 2024) | 33.8 | 1.1 | 2.9 | 17.0 | 0.2 | 26.8 | 1.0 | 4.3 | 53.9 | 5.0 | 20.6 | 2.2 | **64.3** | 27.1 | 53.8 | 27.0 | 37.0 | **28.6** | 8.6 | 21.6 | 23.4 | 27.2 | 21.4 | 21.9 | +6.4 |
| LiDARWeather (Park et al., 2024) | 39.3 | 2.9 | 0.9 | 19.4 | 0.8 | 27.7 | 2.2 | 3.8 | 42.5 | 9.4 | 21.6 | 0.3 | 51.9 | 33.5 | 47.4 | 23.1 | 33.3 | 23.2 | 6.8 | 19.0 | 21.2 | 23.1 | 17.3 | 20.5 | +5.0 |
| NTN (Park et al., 2025) | 48.4 | 1.5 | 2.4 | 19.4 | 0.2 | 29.1 | 3.2 | 8.9 | 43.5 | 6.7 | 20.5 | 0.0 | 52.2 | 30.1 | 49.8 | 20.0 | 32.9 | 24.7 | 7.5 | - | - | - | - | 21.1 | +5.6 |
| A3Point (ours) | **76.7** | **4.0** | **5.0** | **29.6** | **1.3** | **35.1** | 1.7 | **9.5** | **55.4** | 3.9 | **24.0** | **3.5** | 61.7 | **34.5** | **60.1** | **34.1** | 33.3 | 28.1 | **14.8** | **26.8** | **26.6** | **31.9** | **28.6** | **27.2** | +11.7 |

and heavy snow. The extensive augmentation space and adaptive latent-space distillation enable A3Point to handle a wide spectrum of weather disturbances while preserving predictions in less corrupted regions, resulting in better performance than other methods in milder conditions.

**Qualitative Results.** Fig.4 presents a visual comparison of A3Point with previous methods under challenging weather conditions like snow and dense fog. A3Point shows a superior ability to accurately segment major scene components such as *sidewalk*, *road*, and *terrain*, which are often obscured or distorted in adverse weather. Moreover, it exhibits better segmentation of complex instances like *car* and *traffic sign*, which are critical for safe navigation. The local pattern encoding helps capture the unique geometries of these objects, while the decoupling of semantic confusion and shift enhances the discriminative performance of these classes.

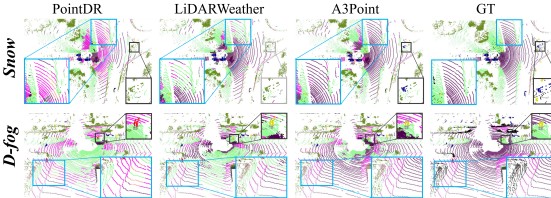

Figure 4: Qualitative results on [A] → [C]. Significant improvements are marked with boxes.

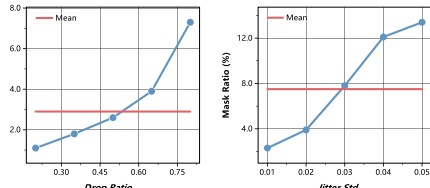

Figure 5: Mask ratio of SSR under different augmentation levels.

## 4.3 ABLATION STUDY

See Appendix F-V for more analyses, discussions, and visualizations.

**Additional Results.** To validate the effectiveness and generalizability of A3Point, we conduct experiments across different architectures and benchmarks (Tab. 3). Results show consistent gains. With SPVCNN, A3Point achieves +12.0%, +8.5%, and +2.2% mIoU on [A] → [C], [B] → [C], and [A] → [D], respectively. Using Minkowski, the improvements are +9.9%, +11.7%, and +5.9% mIoU. Notably, improvements are larger on the real adverse weather benchmark [C] than on the synthetic corruption benchmark [D], indicating superior robustness to real-world disturbances. The consistent gains across architectures highlight the approach's architecture-friendly nature, making it broadly applicable to various LiDAR segmentation networks.

**Effectiveness of Components.** We conduct an ablation study on the [A]/[B]→[C] benchmark to evaluate the impact of A3Point's components (Tab. 4). For [A]→[C], the baseline without augmentations achieves 31.4% mIoU. Introducing the enhanced augmentation space (EAS) improves performance to 38.7%, highlighting the importance of diverse augmentations for domain robustness. Applying semantic shift region localization and masking to only optimize semantic consistency regions (SCR) with $\widetilde{\mathcal{L}}_{ce}$ further improves mIoU to 40.2%. Finally, incorporating latent variable-based

Table 3: Different architectures & benchmarks.

| Method | SPVCNN | | Minkowski | |
|---|---|---|---|---|
| | baseline | w/ A3Point | baseline | w/ A3Point |
| $[A] \rightarrow [C]$ | 28.1 | 40.1 (**+12.0**) | 31.4 | 41.3 (**+9.9**) |
| $[B] \rightarrow [C]$ | 17.3 | 25.8 (**+8.5**) | 15.5 | 27.2 (**+11.7**) |
| $[A] \rightarrow [D]$ | 52.5 | 54.7 (**+2.2**) | 53.0 | 58.9 (**+5.9**) |

Table 4: Ablation study of A3Point components.

| None | EAS ($\hat{\mathcal{L}}_{ce}$) | SCR ($\widetilde{\mathcal{L}}_{ce}$) | SSR ($\mathcal{L}_{distll}$) | $[A] \rightarrow [C]$ | $[B] \rightarrow [C]$ |
|---|---|---|---|---|---|
| ✓ | | | | 31.4 | 15.5 |
| | ✓ | | | 38.7 | 24.9 |
| | ✓ | ✓ | | 40.2 | 26.5 |
| | ✓ | ✓ | ✓ | **41.3** | **27.2** |

Table 5: Ablation study of augmentation level. We define different levels (light, moderate, and heavy), same as Fig.1.

| Method | None | light | moderate | heavy | random |
|---|---|---|---|---|---|
| $[A] \rightarrow [A]$ | 62.7 | 61.2 | 59.7 | 55.5 | 60.4 |
| w/ A3Point | **62.6** | 62.4 | 61.0 | 58.7 | 62.5 |
| $[A] \rightarrow [C]$ | 31.4 | 37.6 | 38.0 | 37.1 | 38.7 |
| w/ A3Point | 31.8 | 38.5 | 40.4 | 40.5 | **41.3** |
| $[B] \rightarrow [C]$ | 15.5 | 19.3 | 23.6 | 24.1 | 24.9 |
| w/ A3Point | 15.6 | 20.7 | 25.1 | 26.4 | **27.2** |

Table 6: Performance comparison of different strategies for modeling semantic confusion prior.

| Strategy | $[A] \rightarrow [C]$ | $[B] \rightarrow [C]$ |
|---|---|---|
| None | 38.7 | 24.9 |
| GT-based | 39.1 | 25.0 |
| Offline | 40.7 | 26.4 |
| Online (ours) | **41.3** | **27.2** |

distillation loss $\mathcal{L}_{distill}$ for semantic shift regions (SSR) achieves the best result of 41.3% mIoU. A similar trend is observed for [B]→[C], where the baseline starts at 15.5% mIoU. EAS significantly boosts performance to 24.9%, bridging the large gap between synthetic and real adverse weather data. Adding SCR improves mIoU to 26.5%, and the full model with SSR reaches 27.2%. These results demonstrate the complementary nature of our components across domain generalization scenarios. By leveraging learned semantic confusion priors to guide semantic shift optimization, the model adapts to novel disturbances while preserving semantic consistency.

**Analysis of Augmentation Level.** We analyze the impact of augmentation levels on network performance using the [A] validation set ([A]→[A]) and cross-domain benchmarks [A]/[B]→[C] (Tab. 5). On [A]→[A], baseline performance drops from 62.7% to 55.5% mIoU as augmentation increases from none to heavy, showing the adverse effect of severe augmentations. In contrast, A3Point maintains higher performance, with only a slight decrease from 62.6% to 58.7%, demonstrating its ability to mitigate the negative impact of aggressive augmentations by handling semantic shift regions. For [A]→[C], augmentations improve baseline performance from 31.4% to 38.7% mIoU, confirming their role in enhancing robustness. However, a slight drop from 38.0% to 37.1% from moderate to heavy augmentations suggests emerging semantic shift issues. A3Point avoids this drop, achieving 40.5% mIoU with heavy augmentations and 41.3% mIoU with random augmentations. Similar trends appear in [B]→[C], where A3Point achieves larger relative gains as augmentation intensity increases. The performance gap over the baseline grows from 1.4% with light augmentations to 2.3% with random augmentations. These findings confirm that A3Point effectively handles semantic shift, allowing the use of aggressive augmentations without harming performance. Its ability to balance source and target performance makes it well-suited for real-world applications.

**Analysis of Semantic Confusion Prior.** Tab. 6 compares different strategies for modeling semantic confusion prior on [A]/[B]→[C] benchmarks. For [A]→[C], the GT-based method, using one-hot ground truth labels, models only class-wise shape distribution but lacks inter-class confusion knowledge, yielding a marginal 0.4% mIoU gain. The offline method, which uses a pre-trained model's predictions on the source domain, improves mIoU by 2.0% but struggles to capture evolving confusion patterns during training. Similar trends occur in [B]→[C], where the GT-based method provides minimal gains (25.0% vs. 24.9%), and the offline method achieves better but limited improvement (26.4% vs. 24.9%). In contrast, our online modeling approach continuously updates the prior information, effectively decoupling semantic confusion in augmented point clouds. This dynamic strategy achieves the best performance: 41.3% mIoU on [A]→[C] and 27.2% mIoU on [B]→[C], outperforming the baseline by 2.6% and 2.3%, respectively. By continuously adapting to evolving confusion patterns, A3point mitigates semantic shifts and consistently outperforms static priors, demonstrating the importance of dynamic prior modeling in our framework.

**Analysis of Semantic Shift Region.** Fig.5 shows how the SSR mask ratio varies with augmentation level. As the augmentation intensity increases, the mask ratio of the localized SSR grows accordingly. This observation aligns with our expectation that stronger perturbations lead to more significant semantic shift in the augmented point clouds, resulting in a larger proportion of the input being identified as belonging to the SSR. The upward trend in the curve suggests that our proposed SSR localization module effectively captures the regions most affected by the augmentation-induced

semantic shift, enabling the network to apply appropriate optimization strategies in these areas to mitigate the impact of the shift and enhance the model's robustness to adverse weather conditions.

## 5 CONCLUSION

This paper presents A3Point, an adaptive augmentation-aware latent learning framework for robust LiDAR semantic segmentation under adverse weather. A3Point effectively leverages a large augmentation space while mitigating semantic shift through a two-step strategy: (1) semantic confusion prior latent learning, which encodes local confusion patterns through discrete latent representations, and (2) semantic shift region localization, which detects anomalies in augmented point clouds to separate semantic consistency from semantic shift regions, enabling targeted optimization. Experiments on domain generalization benchmarks demonstrate its effectiveness, particularly in generalizing from synthetic normal weather to real adverse weather.

## ACKNOWLEDGEMENTS

This work was partially supported by the National Key R&D Program of China (Grant No. 2024YFB3909902), National Natural Science Foundation of China (Grant No. U25A20536), and the Youth Innovation Promotion Association of the Chinese Academy of Sciences (CAS).

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

## A  APPENDIX

This appendix provides additional details and analyses to complement our main paper. We include implementation details, ablation studies, and additional visualization results. Specifically, the appendix is organized as follows:

- **Section B** Implementation: Semantic Confusion Prior (SCP) discrete latent learning (VQ-VAE design, per-class quantization, losses, and reconstruction).

- **Section C** Implementation: Semantic Shift Region (SSR) localization and anomaly detection strategy, mask generation, and dilation.

- **Section D** Discussion: Discrete vs. continuous encodings—why discrete representations better match our prior and interpretability goals.

- **Section E** Algorithm: A3Point training pseudocode and end-to-end optimization pipeline.

- **Section F** Augmentation: Impact analysis of additional data augmentation techniques and conclusions.

- **Section G** Hyperparameters: Sensitivity and trade-offs of key A3Point hyperparameters $(k, D, t, \lambda)$.

- **Section H** Visualization: SSR visual analysis and association with error regions.

- **Section I** Visualization: More qualitative results.

- **Section J** Evaluation: Training-time SSR-region evaluation and inference-time high-distortion subregion evaluation.

- **Section K** Training overhead: Extra cost during training and zero-overhead inference statement.

- **Section L** Comparison: Feasibility of replacing VQ-VAE with prototypes (K-prototype) and sources of performance gaps.

- **Section M** Schedule: Comparison between 50-epoch and 15-epoch training schedules and their effects.

- **Section N** Confusion: Things-class confusion matrix and frequency–gain relationship analysis.

- **Section O** Universality: Commonality of semantic confusion across datasets/architectures and the role of online priors.

- **Section P** Rare classes: Oracle's 0.0 IoU phenomenon on extremely rare classes and A3Point's stabilizing effect.

- **Section Q** EAS: "Safe training" capability under larger augmentation spaces and its upper bounds.

- **Section R** Iterative: Curriculum-style augmentation ceilings and iterative A3Point.

- **Section S** Fairness: Comparisons after equipping prior methods with stronger augmentations.

- **Section T** Codebook: t-SNE visualization and interpretation of per-class VQ-VAE sub-codebooks.

- **Section U** Method comparison: Controlled comparisons with Mean Teacher, SCE, etc., under the same augmentation space.

- **Section V** Design choice: Ablation verifying why we use "global nearest code" for distillation supervision within SSR.

- **Section W** Class mapping across datasets: Unified 19-class protocol and exact mappings for [A]/[B]/[C].

- **Section X** Impact: LLM usage notes and societal impact discussion.

# B   IMPLEMENTATION OF SCP LATENT LEARNING

In this section, we provide the implementation details of our semantic confusion prior (SCP) latent learning module, which is based on the vector quantized variational autoencoder (VQ-VAE) architecture (Van Den Oord et al., 2017). The goal is to learn a discrete latent representation that captures the semantic confusion patterns in the original point cloud predictions.

## B.1   VQ-VAE ARCHITECTURE

The VQ-VAE consists of an encoder, a decoder, and a vector quantization layer. The encoder network maps the input point cloud features $x$ concatenated with the predicted probabilities $f(x)$ to a latent representation $z_e$. We use sparse 3D convolutions (Tang et al., 2020) in the encoder to process the sparse point cloud data efficiently. The architecture is as follows:

- The encoder has 4 sparse conv3d downsampling blocks with channel dimensions [16, 32, 64, 128]. Each block consists of a stride-2 sparse conv3d, batch norm, and LeakyReLU. This downsamples the point cloud by 16x.
- An additional sparse conv3d, batch norm and LeakyReLU, followed by 2 residual blocks to further process the features.
- A final sparse conv3d to map to the latent dimension (64).

The decoder network reconstructs the predicted probabilities $f(x)$ from the quantized latent representation $z_q$. It follows a mirrored architecture to the encoder:

- A sparse conv3d, batch norm and LeakyReLU to map from the latent dimension to the initial feature dimension (128).
- 2 residual blocks for further processing.
- 4 sparse transposed conv3d upsampling blocks, each consists of a stride-2 transposed sparse conv3d, batch norm and LeakyReLU. This upsamples the features by 16x back to the original resolution.
- A final sparse conv3d and tanh activation to reconstruct the predicted probabilities.

## B.2   PER-CLASS VECTOR QUANTIZATION

To avoid inter-class confusion, we perform vector quantization independently for each semantic class. The VQ codebook $\mathcal{C}$ contains learnable embeddings for each class, with dimensions $C \times k \times D$, where $C$ is the number of classes, $k$ is the number of embeddings per class, and $D$ is the latent dimension.

During the forward pass, we split the encoder output $z_e$ into class-specific latents based on the class labels. For each class, the latents are quantized to the nearest embedding in its corresponding sub-codebook using L2 distance. The distances to embeddings of other classes are masked out to prevent selecting incorrect classes.

## B.3   COMMITMENT LOSS AND EMBEDDING LOSS

The learning objective combines a commitment loss and an embedding loss, following the original VQ-VAE:

- The commitment loss encourages the encoder output $z_e$ to commit to the selected embeddings $z_q$, and is defined as:
$$\mathcal{L}_{\text{commit}} = \|z_e - \text{sg}(z_q)\|_2^2 \tag{12}$$
- The embedding loss brings the selected embeddings $z_q$ close to the encoder output $z_e$, and is defined as:
$$\mathcal{L}_{\text{embed}} = \|\text{sg}(z_e) - z_q\|_2^2 \tag{13}$$

where sg stands for the stop-gradient operation. The total VQ loss is:

$$\mathcal{L}_{\text{VQ}} = \mathcal{L}_{\text{commit}} + \beta \mathcal{L}_{\text{embed}} \tag{14}$$

where $\beta$ is a weighing coefficient (set to 0.25).

### B.4 RECONSTRUCTION LOSS

To ensure the quantized representation can reconstruct the input predicted probabilities, we employ a mean squared error reconstruction loss:

$$\mathcal{L}_{\text{recon}} = \|f(x) - \mathcal{D}(z_q)\|_2^2 \tag{15}$$

The final loss is a weighted combination of the reconstruction loss and the VQ losses:

$$\mathcal{L} = \mathcal{L}_{\text{recon}} + \mathcal{L}_{\text{VQ}} \tag{16}$$

In summary, our SCP latent learning module utilizes a VQ-VAE architecture with sparse convolutions to learn a discrete latent representation of semantic confusion patterns in a class-wise manner. The model is trained end-to-end using a combination of reconstruction loss and VQ losses. This allows capturing informative priors on the semantic confusion from the original point cloud predictions.

## C IMPLEMENTATION OF SSR LOCALIZATION

In this section, we describe the implementation details of our semantic shift region (SSR) localization module. The goal is to identify regions in the augmented point cloud where the network's predictions deviate from the learned semantic confusion priors, indicating potential semantic shifts caused by the augmentations.

### C.1 TRACKING LATENT REPRESENTATION DISTRIBUTIONS

During the training of the SCP latent learning module, we track the distribution of latent representations associated with each quantized latent variable (i.e., each embedding in the codebook). Specifically, for each latent variable $e_i$, we calculate the variance $\sigma_i^2$ of its corresponding latent representations before quantization. The variance is updated using an exponential moving average (EMA) with a momentum factor $\gamma$ (set to 0.9):

$$\sigma_i^2 \leftarrow \gamma \sigma_i^2 + (1 - \gamma)\text{Var}(z_e | z_q = e_i) \tag{17}$$

where $\text{Var}(\cdot)$ denotes the variance operation. This allows us to estimate the typical distribution of latent representations for each latent variable.

### C.2 LOCATING SEMANTIC SHIFT REGIONS

To locate the semantic shift regions, we first pass the augmented point cloud through the trained encoder $\mathbb{E}$ to obtain its latent representations $z_e$. We then compare each latent representation to the distribution of its nearest latent variable in the corresponding sub-codebook.

Specifically, for each latent representation $z_e^j$, we find its nearest latent variable $e_{NN(j)}$ in the sub-codebook of the corresponding class. We consider $z_e^j$ to be an outlier (i.e., belonging to a semantic shift region) if its distance from $e_{NN(j)}$ exceeds a threshold based on the tracked variance $\sigma_{NN(j)}^2$:

$$\text{isOutlier}(z_e^j) = \mathbb{1}\left(\|z_e^j - e_{NN(j)}\|_2 > t\sqrt{\sigma_{NN(j)}^2}\right) \tag{18}$$

where $\mathbb{1}(\cdot)$ is the indicator function and $t$ is a hyperparameter controlling the threshold (set to 3). The intuition is that if a latent representation deviates significantly from the typical distribution of its nearest latent variable, it likely corresponds to a semantic shift region.

## C.3 GENERATING SSR AND SCR MASKS

After identifying the outlier latent representations, we project them back to the point cloud space to generate masks for the semantic shift regions (SSR) and semantic consistency regions (SCR).

We first create a binary mask $M_{\text{outlier}}$ in the latent representation space, where $M_{\text{outlier}}^j = \text{isOutlier}(z_e^j)$. We then use the transpose of the point cloud downsampling operation (used in the encoder) to upsample the mask back to the original point cloud resolution. This gives us the SSR mask $M_{\text{SSR}}$.

The SCR mask is simply the complement of the SSR mask:

$$M_{\text{SCR}} = \mathbb{1} - M_{\text{SSR}} \tag{19}$$

## C.4 HANDLING SPARSE OUTLIERS

In practice, the outlier latent representations may be sparse and scattered. To ensure the SSR mask covers semantically meaningful regions, we perform a dilation operation on the SSR mask. Specifically, for each point $(x, y, z, c)$ in the SSR mask, we set its neighboring points within a certain radius $r$ to also belong to the SSR. This helps to connect nearby outlier points and form contiguous semantic shift regions.

The dilation operation can be efficiently implemented by first upsampling the outlier mask to a dense 3D grid, performing dilation on the grid, and then downsampling the dilated mask back to the point cloud using nearest-neighbor interpolation.

In summary, our SSR localization module identifies semantic shift regions in the augmented point cloud by comparing the latent representations to the learned semantic confusion priors. It generates SSR and SCR masks to guide the subsequent training with appropriate losses for each region. The module is computationally efficient and can be integrated into the training pipeline without significantly increasing the training time.

## D    DISCUSSION: DISCRETE VS. CONTINUOUS ENCODING

A key aspect of the proposed Semantic Confusion Prior (SCP) is **explicitly obtaining the distribution form of the representation** for subsequent localization and optimization. This section discusses the differences between discrete and continuous encoding methods and explains why discrete encoding is more suitable for our framework.

## D.1    COMPARISON OF ENCODING PARADIGMS

**Discrete Encoding (e.g., VQ-VAE).** VQ-VAE (Van Den Oord et al., 2017) learns a discrete latent space by: 1) Mapping the input to a latent space via an encoder. 2) Quantizing the latent space into a codebook, effectively clustering representative features.

**Continuous Encoding Methods.** Continuous representation methods can be categorized into two types: **(1) Fixed Prior Distribution Paradigms (e.g., VAE (Kingma et al., 2013), Flow-based Models (Rezende & Mohamed, 2015; Kingma & Dhariwal, 2018))** These methods constrain the latent space to a fixed prior distribution (e.g., Gaussian) to achieve implicit global distribution alignment. However, this alignment lacks interpretability and cannot naturally decouple different semantic classes, often requiring separate encoders and decoders per class. **(2) Unconstrained Latent Representations (e.g., AE (Rumelhart et al., 1986), DAE (Vincent et al., 2008), SDAE (Vincent et al., 2010))** These methods learn continuous latent representations without a predefined prior, making it difficult to directly extract structured distribution information. As a result, clustering techniques may still be needed to impose structure, similar to VQ-VAE's discrete encoding.

---

**Algorithm 1** Pseudo Algorithm of A3Point

---

1: **Inputs:** Source domain $D_S = \{(x_i^S, y_i^S)\}_{i=1}^{N_S}$
2: **Define:** Network $f_\theta$, Autoencoder $\mathbb{E}, \mathbb{D}$, Codebook $\mathcal{C}$, Augmentation function $\mathcal{A}$, Learning rates $\alpha, \beta, \delta$, Loss weight $\lambda$
3: **Output:** Trained model $f_\theta$
4: **for** each batch $(x_i^S, y_i^S)$ in $D_S$ **do**
5:     *# Step 1: Augmented Training*
6:     Apply augmentations: $(\hat{x}_i^S, \hat{y}_i^S) = \mathcal{A}(x_i^S, y_i^S)$
7:     Compute supervised loss: $\mathcal{L}_{ce} = \ell_{ce}(f_\theta(x_i^S), y_i^S)$         ▷ Eq. (1)
8:     Compute augmented loss: $\hat{\mathcal{L}}_{ce} = \ell_{ce}(f_\theta(\hat{x}_i^S), \hat{y}_i^S)$         ▷ Eq. (2)
9:     *# Step 2: Semantic Confusion Prior Learning*
10:     Compute latent embedding: $z_e = \mathbb{E}([f_\theta(x_i^S) \oplus x_i^S])$
11:     Quantize latent code: $z_q = \text{quantize}(z_e)$ using codebook $\mathcal{C}$         ▷ Eq. (3)
12:     Reconstruct prediction: $\bar{x}_i^S = \mathbb{D}(z_q)$
13:     Update VQ-VAE loss: $\mathcal{L}_{vq} = ||x_i^S - \bar{x}_i^S||_2^2 + ||\text{sg}(z_e) - z_q||_2^2 + ||z_e - \text{sg}(z_q)||_2^2$  ▷ Eq. (4)
14:     *# Step 3: Semantic Shift Region Localization*
15:     Track variance statistics for latent codes $\sigma^2$ using EMA
16:     Map augmented predictions to latent space: $z_e^{aug} = \mathbb{E}([f_\theta(\hat{x}_i^S) \oplus \hat{x}_i^S])$
17:     Identify semantic shift regions (SSR) via anomaly detection:
18:       - Semantic consistency mask: $M_{SCR} = \mathbb{1}(z_e^{aug} \in r(NN_{sub}(z_e^{aug})))$     ▷ Eq. (7)
19:       - Semantic shift mask: $M_{SSR} = \mathbb{1}(z_e^{aug} \notin r(NN_{sub}(z_e^{aug})))$     ▷ Eq. (8)
20:     *# Step 4: Optimization*
21:     Compute loss for SCR: $\widetilde{\mathcal{L}}_{ce} = \ell_{ce}(f_\theta(\hat{x}_i^S) \odot M_{SCR}, y_i^S \odot M_{SCR})$     ▷ Eq. (9)
22:     Compute distillation loss for SSR: $\mathcal{L}_{distill} = ||z_e^{aug} - \text{sg}[NN_{global}(z_e^{aug})]||_2^2$  ▷ Eq. (10)
23:     Update model: $\theta \leftarrow \theta - \delta \nabla_\theta (\mathcal{L}_{ce} + \widetilde{\mathcal{L}}_{ce} + \lambda \mathcal{L}_{distill})$     ▷ Eq. (11)
24: **end for**

---

### D.2 WHY DISCRETE ENCODING IS NECESSARY?

While discrete encoding limits the number of latent patterns to $k$, this does not hinder our framework's effectiveness. Instead, it provides several advantages:

**Better Interpretability:** Each latent code represents a specific semantic confusion pattern, making it easier to analyze and interpret the learned representations.

**Class-wise Separation:** VQ-VAE allows natural clustering of latent representations per class, which is crucial for our semantic shift localization.

**Robustness and Generalization:** A well-designed codebook ensures that only the most representative and meaningful patterns are learned, improving generalization to unseen data.

## E PSEUDO ALGORITHM OF A3POINT.

We further provide detailed algorithmic description in Alg. 1.

## F IMPACT OF OTHER AUGMENTATION TECHNIQUES

To further analyze the role of various augmentation techniques, we conduct experiments to evaluate their impact on both source and target domain performance.

### F.1 EXPERIMENTAL SETUP

We follow the same training setup as in Section 3.3.2, varying the intensity of secondary augmentations such as scaling, flipping, and noise perturbation, while keeping point deformation and point loss augmentations unchanged. We define three augmentation levels: - **Light**: Default augmentation settings used in our main experiments. - **Moderate**: Increasing the magnitude of scaling and flip-

ping while maintaining the original distribution. - **Heavy**: Aggressively increasing scaling factors and noise perturbation levels.

## F.2 RESULTS AND ANALYSIS

Tab. 7 presents the results on the [A]→[C] and [B]→[C] benchmarks.

| Augmentation Level | [A]→[C] | [B]→[C] | [A]→[A] |
|---|---|---|---|
| Light | **41.3** | **27.2** | **62.5** |
| Moderate | 40.1 | 26.0 | 60.3 |
| Heavy | 37.5 | 24.5 | 55.8 |

Table 7: Impact of different augmentation levels on domain generalization performance.

From the results, we observe that: 1. **Excessive secondary augmentations degrade both source and target performance.** Increasing the augmentation magnitude from Mild to Heavy reduces mIoU by 3.8% on [A]→[C] and 2.7% on [B]→[C]. 2. **Moderate augmentations provide limited benefit.** Compared to the Mild setting, Moderate augmentation slightly reduces performance, indicating that secondary augmentations do not contribute significantly to domain adaptation. 3. **Source domain performance deteriorates with stronger augmentations.** On [A]→[A], mIoU drops significantly from 62.6% to 55.8%, suggesting that excessive scaling and noise perturbation disrupt the original data distribution and hinder learning.

These findings confirm that point deformation and loss (jittering, point drop) are the primary drivers of domain adaptation, while other augmentations are actually domain-agnostic factors and should be applied conservatively to avoid performance degradation.

## G INFLUENCE OF PARAMETERS SETTING

In this section, we discuss the key hyperparameters of our A3Point framework and their impact on the performance and behavior of the model. All experiments are conducted on SemanticKITTI→ semanticSTF.

### G.1 LATENT SPACE DIMENSIONS

The dimensions of the latent space in the SCP latent learning module, determined by the number of embeddings per class ($k$) and the embedding dimension ($D$), affect the expressiveness and granularity of the learned semantic confusion patterns.

| $k$ | $D$ | mIoU (%) |
|---|---|---|
| 16 | 64 | 39.9 |
| 32 | 64 | 41.3 |
| 64 | 64 | 41.4 |
| 32 | 32 | 40.8 |
| 32 | 64 | 41.3 |
| 32 | 128 | 41.2 |

Table 8: Ablation study on latent space dimensions.

In our implementation, we set $k = 32$ and $D = 64$ (Table 8). These values provide a good trade-off between the representational power of the latent space and the computational efficiency of the module. Increasing $k$ allows the model to capture more diverse semantic confusion patterns within each class, but it also increases the memory footprint and the risk of overfitting. Similarly, increasing $D$ enhances the capacity of each embedding to encode more complex patterns but also increases the computational overhead.

## G.2 SSR LOCALIZATION THRESHOLD

The threshold $t$ used in the SSR localization module determines the sensitivity of detecting semantic shift regions.

| | $t = 2$ | $t = 3$ | $t = 4$ |
|---|---|---|---|
| mIoU (%) | 41.0 | 41.3 | 40.9 |

Table 9: Ablation study on SSR localization threshold.

In our experiments, we set $t = 3$, which effectively identifies the regions with substantial semantic shifts while minimizing false positives. A higher value of $t$ results in a more conservative approach, where only the most significant deviations from the learned semantic confusion patterns are considered as semantic shifts. Conversely, a lower value of $t$ makes the module more sensitive, potentially identifying more regions as semantic shifts.

## G.3 LOSS WEIGHTING COEFFICIENT

The loss weighting coefficient $\lambda$ balances the contributions of the cross-entropy loss and the distillation loss in the overall training objective.

| | $\lambda = 0.02$ | $\lambda = 0.1$ | $\lambda = 0.5$ |
|---|---|---|---|
| mIoU (%) | 40.5 | 41.3 | 40.9 |

Table 10: Ablation study on loss weighting coefficient.

In our experiments, we found that setting $\lambda = 0.1$ achieves a good balance between the two loss terms. A higher value of $\lambda$ gives more importance to the distillation loss, encouraging the model to focus more on aligning the predictions in the semantic shift regions with the learned semantic confusion patterns. On the other hand, a lower value of $\lambda$ emphasizes the cross-entropy loss, prioritizing the overall segmentation accuracy.

## H VISUAL ANALYSIS OF SSR

To better understand how our method identifies semantic shift regions (SSR), we conduct a detailed visual analysis comparing the error maps with the detected SSR, as shown in Fig. 6. The error map highlights regions where the model's predictions differ from ground truth labels, while SSR indicates areas identified by our semantic shift detection mechanism.

From the visualization results, we observe a strong correlation between the SSR and regions prone to prediction errors. Specifically: (1) Our SSR detection effectively captures areas where adverse weather conditions cause significant semantic ambiguity, particularly at object boundaries and distant regions where point cloud density decreases. (2) The SSR often corresponds to challenging scenarios such as intersections between different semantic categories (e.g., road-sidewalk boundaries) and areas with complex geometric structures. (3) The semantic consistency regions (non-SSR areas) generally align well with regions where the model maintains accurate predictions, validating our approach's ability to identify reliable predictions. This visual analysis demonstrates that our SSR detection mechanism provides meaningful guidance for applying different optimization strategies during training.

## I MORE QUALITATIVE RESULTS

To further demonstrate the effectiveness of our proposed method, we present additional qualitative results comparing our approach with baseline methods on the challenging SemanticKITTI→SemanticSTF domain generalization task, as shown in Fig. 7. These qualitative results further validate the effectiveness of our semantic confusion prior learning and semantic shift region localization strategies in improving domain generalization performance for LiDAR semantic segmentation.

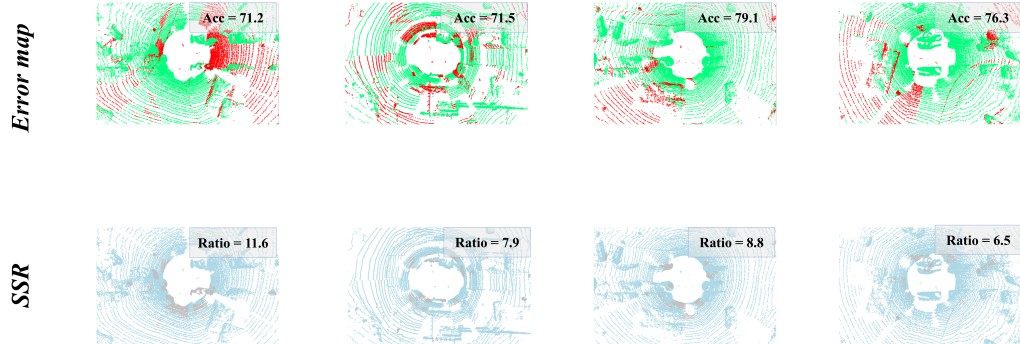

Figure 6: Qualitative results on comparison between error map with semantic shift region.

## J  TARGETED EVALUATIONS IN SSR AND HIGH-DISTORTION REGIONS

Appendix H visualizes the correlation between SSR and error maps, showing substantial overlap with error-prone areas. To more directly quantify improvements, we add two complementary evaluations:

- **Training-time SSR-region evaluation (approximate mIoU):** We use training-time samples and their SSR masks to generate fixed evaluation masks and track mIoU changes within SSR-region:
    - Generate a fixed evaluation mask $\mathcal{M}_{\text{SSR\_eval}}$ using the same rules (we use the prior encoder and variance statistics obtained from the EAS-only setting).
    - Because GT in SSR is unreliable (by definition, labels are misaligned after strong augmentation), we follow semi-supervised practice and maintain an EMA (mean-teacher) model as a proxy evaluator. EMA is preferable to GT here because the EMA provides a stable, denoised target that tracks the model's long-term belief without inheriting the semantic misalignment introduced by strong augmentations. Better agreement with the EMA in SSR indicates improved consistency and reliability precisely where GT is noisy.

  As shown in Fig.8, compared to EAS-only, A3Point's predictions within $\mathcal{M}_{\text{SSR\_eval}}$ exhibit higher agreement with the EMA teacher. In particular, adding SSR distillation drives predictions to align more closely with the teacher within SSR regions.

- **Inference-time high-distortion subregions evaluation:** As shown in Fig.9, we approximate "high-distortion" subregions using density/curvature thresholds. In these subregions, A3Point improves the top-5 frequent classes by 1.9–15.1 points and raises average IoU by 7.6 over EAS. Construction details:
    - For each point $i$, find $k$-NN $N_k(i)$ with $k = 32$.
    - Local $\text{density}(i) \approx$ inverse of the distance to the $k$-th nearest neighbor.
    - Local curvature via PCA on $N_k(i)$: compute the covariance matrix $\Sigma_i$, obtain eigenvalues $\lambda_1 \geq \lambda_2 \geq \lambda_3 \geq 0$, define $\text{curvature}(i) = \lambda_3/(\lambda_1 + \lambda_2 + \lambda_3)$.
    - Aggregate point-wise density/curvature to voxels $v$ via mean to obtain $\text{density}(v)$ and $\text{curvature}(v)$.
    - Define the high-distortion indicator using quantile-based thresholds (selecting 20% hardest regions):

    $$\text{HighDist}(i) = \mathbb{I}\left(\text{density}(i) \leq \tau_d \ \vee \ \text{curvature}(i) \geq \tau_c\right).$$

    - Within these high-distortion masks, A3Point outperforms EAS, as shown in Table 11.

These analyses support that our method not only improves overall metrics but also meaningfully enhances performance within the high-risk regions detected during training.

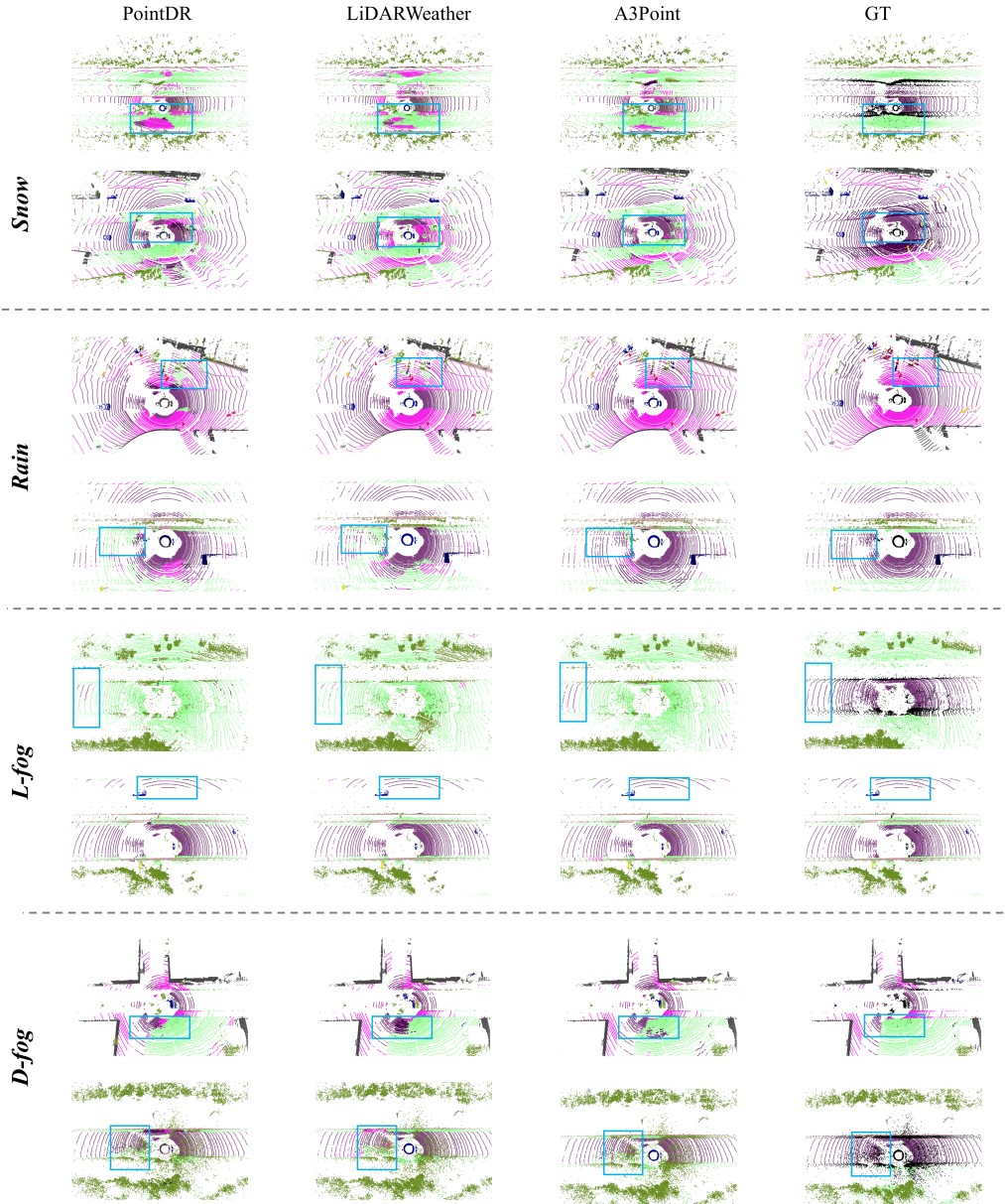

Figure 7: Qualitative results on [A] → [C], where improvements are marked with boxes.

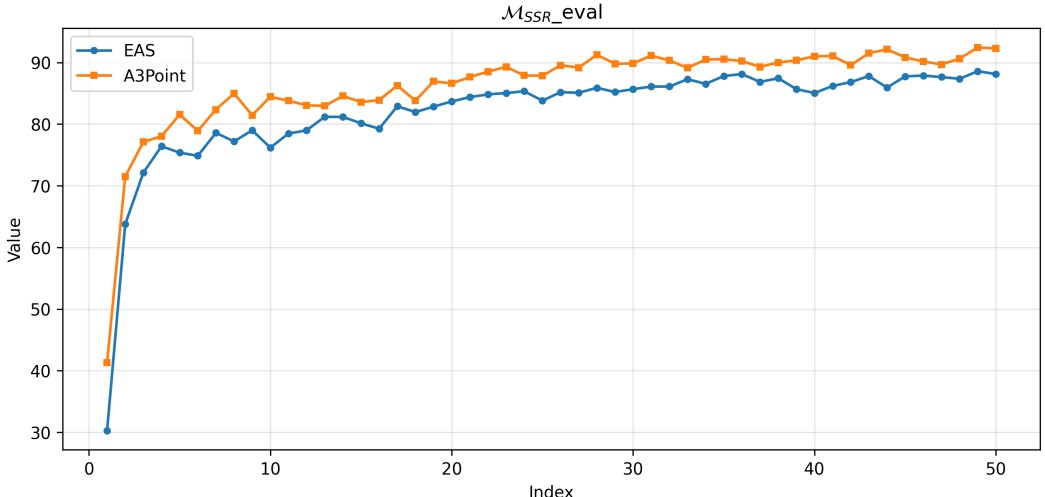

Figure 8: Agreement with EMA teacher within $\mathcal{M}_{\text{SSR\_eval}}$.

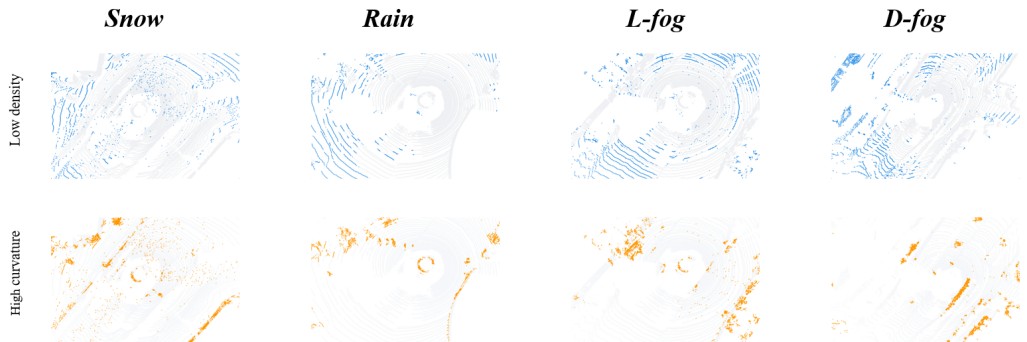

Figure 9: Example scenes showing "high-distortion" subregions.

Table 11: Performance within high-distortion subregions.

| Method | Car | Road | Sidewalk | Vegetation | Terrain | Overall mIoU |
|---|---|---|---|---|---|---|
| EAS | 61.6 | 50.7 | 21.4 | 47.5 | 35.7 | 27.1 |
| A3Point | 76.7 | 57.6 | 26.8 | 51.5 | 37.6 | 34.7 |

## K  TRAINING-TIME OVERHEAD

A3Point introduces a lightweight, class-wise VQ-VAE module—implemented with sparse convolutions, a code dimension of $D = 64$, and a per-class codebook size of $k = 32$ (see architecture details in Appendix B.1)—to enable SCP and SSR during training. Below we detail the practical implications and costs:

- **No inference-time overhead.** Both SCP and SSR are training-only components. They are disabled at test time, so inference latency and memory footprint are identical to the baseline model. The exported checkpoint and forward path remain unchanged.

- **Low-cost statistics via EMA.** We maintain per-codeword statistics (variance) using an exponential moving average (EMA). This update is $O(1)$ per iteration with negligible extra memory and does not grow with dataset size or batch length.

- **Frozen prior for SSR localization.** The encoder used to localize SSR regions is kept frozen during localization and does not backpropagate. This design bounds the incremental compute to a lightweight forward pass and avoids gradient overhead, keeping utilization stable.

To show the cost clearly, we report a representative runtime snapshot under identical hardware and training settings in Table 12. Given the substantial mIoU gains on SemanticSTF ( +9.9 / +11.7 over the baseline in [A]→[C] and [B]→[C]), we regard the training-time increase as a favorable trade: concentrate budget on the training side to obtain stronger adverse-weather robustness, while keeping the inference path clean and unchanged.

| Method | GPU Memory (MB) | Iterations/s (it/s) | Total Time (h) |
|---|---|---|---|
| Baseline | 8,496 | 2.72 | 24.4 |
| PointDR | 9,624 | 1.97 | 33.7 |
| EAS | 8,932 | 2.05 | 32.4 |
| A3Point (ours) | 17,144 | 1.61 | 41.3 |

Table 12: Training-time overhead comparison under identical hardware and settings.

## L  ON REPLACING VQ-VAE WITH A PROTOTYPE-BASED ALTERNATIVE

We evaluate whether a simple prototype-based approach can replace the VQ-VAE prior. Specifically, we implement a class-wise K-prototype variant ($k = 32$) as a drop-in alternative to the VQ-VAE module. Under the same training budget and augmentation settings, VQ-VAE consistently performs better, as summarized in Table 13.

| Setting | [A]→[C] | [B]→[C] |
|---|---|---|
| None | 31.4 | 15.5 |
| EAS | 38.7 | 24.9 |
| Prototype-based | 40.0 | 25.4 |
| VQ-VAE-based (ours) | **41.3** | **27.2** |

Table 13: Comparison of class-wise K-prototype versus VQ-VAE prior under identical settings.

We attribute the gap to three advantages of the VQ-VAE prior:

- **Intra-class multimodality.** Error/ambiguity patterns within a class are inherently multimodal (e.g., lane markings near curbs, partial occlusions, far-range sparsity). Per-class codebooks learned by VQ-VAE capture multiple modes more faithfully than a single prototype centroid, yielding more accurate localization of risky subregions.

- **Well-formed anomaly modeling.** VQ-VAE supports per-codeword variance tracking (via EMA), producing compact, approximately Gaussian local clusters and an interpretable "prior radius" for anomaly detection. Prototype methods typically require additional metric-learning losses or finely tuned thresholds to behave well and can be brittle across conditions.

- **Online adaptivity and stability.** Our prior is updated online in sync with the evolving backbone while the SSR-localization encoder remains frozen for backprop, providing stable yet responsive statistics. Prototype updates can lag or drift as the feature space shifts, degrading SSR localization quality over training.

In short, VQ-VAE offers a more robust, interpretable, and empirically stronger mechanism for modeling class-wise priors, delivering consistent gains at only modest additional training cost.

## M    TRAINING SCHEDULE: 50 EPOCHS VS. LIDARWEATHER'S 15 EPOCHS

We follow a PointDR-like schedule of 50 epochs for all main results and ablations (Tables 3–6) to ensure internal fairness. To disentangle schedule length from algorithmic gains, we conduct controlled comparisons under both short (15-epoch, LiDARWeather-style) and long (50-epoch) schedules.

**Short schedule (15 epochs)** Under a 15-epoch schedule (Table 14), A3Point still surpasses our LiDARWeather reproduction by +2.7–4.5 mIoU with identical training budgets, indicating that our gains are not contingent on longer training.

| Method | [A]→[C] | [B]→[C] |
|---|---|---|
| Baseline | 31.4 | 15.5 |
| LiDARWeather* | 37.8 | 20.4 |
| A3Point (ours) | **40.5** | **25.9** |

Table 14: 15-epoch (LiDARWeather-style) training. * denotes the reproduced result.

**Long schedule (50 epochs)** Extending Baseline and LiDARWeather to 50 epochs (Table 15) yields only modest improvements and both remain below A3Point, suggesting that schedule length alone does not close the gap.

| Method | [A]→[C] | [B]→[C] |
|---|---|---|
| Baseline* | 32.3 | 16.1 |
| LiDARWeather* | 38.6 | 21.6 |
| A3Point (ours) | **41.3** | **27.2** |

Table 15: 50-epoch training. * denotes the reproduced result.

## N    CONFUSION MATRIX FOR "THINGS" CLASSES

We include confusion matrices and per-class point counts for both the Oracle and A3Point on SemanticSTF in Fig.10 to clarify how class frequency relates to performance gains. Our key observations are as follows:

- **Frequency is not a sufficient predictor of gains.** Classes such as *person* and *motorcyclist*, despite nontrivial aggregate frequency, frequently suffer from severe occlusions,

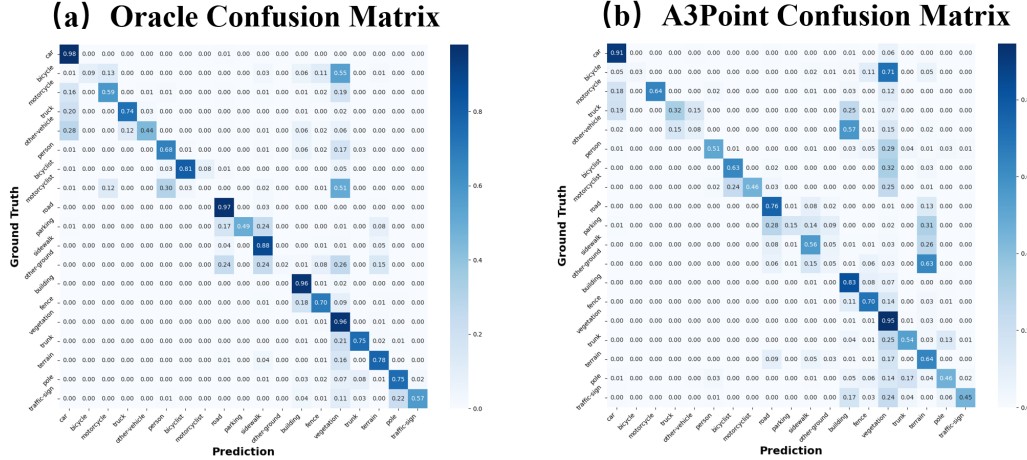

Figure 10: Confusion matrices on SemanticSTF. Left: Oracle. Right: A3Point.

pose/shape variability, and far-range sparsity. Under adverse weather, these factors exacerbate inter-class confusions (e.g., *person* vs. *pole/sign*, *motorcyclist* vs. *bicycle/motorcycle*), effectively reducing the number of reliable points per scene and limiting the attainable improvements.

- **Larger gains for classes with distinctive, stable geometry.** Categories such as *bicycle*, *motorcycle*, *traffic sign*, and *car* preserve clearer geometric signatures under perturbations (e.g., edges, planar surfaces, and repeatable part configurations). Our SCP/SSR priors help maintain consistency in these regions, leading to stronger improvements, consistent with the trends reported in Tables 1-2.

Overall, the confusion matrices reveal that robustness gains are better explained by the stability and distinctiveness of local geometric cues under weather shifts, rather than by raw class frequency alone.

## O    UNIVERSALITY OF SEMANTIC CONFUSION

This section clarifies how "semantic confusion" manifests across datasets and architectures with three complementary points:

1. **Dataset-driven patterns vs. model-driven intensity.** Across architectures, which classes tend to be confused is largely governed by dataset geometry, density, and sampling statistics (e.g., *road–sidewalk*, *car–truck*). By contrast, the magnitude of confusion (confidence spread, error rates) depends more on the model's capacity, regularization, and training dynamics. In our setup, SCP learns confusion priors online on the source domain for each backbone; for a fixed backbone, the confusion *pattern* remains stable while its *intensity* shrinks as the main network improves.

2. **Architectural alignment via online priors.** A3Point learns class-wise discrete priors from the current model's predictions on the original domain (via VQ-VAE) and freezes them for anomaly detection on augmented samples. This "online fit, frozen detect" design keeps the priors aligned with the model's evolving decision boundary and avoids cross-architecture mismatch.

3. **Empirical evidence across backbones and settings.** Table 3 shows consistent gains on two voxel backbones (MinkowskiNet-18/32w and SPVCNN) and three DG tracks ([A]→[C], [B]→[C], [A]→[D]). While absolute confusion varies with the backbone, A3Point consistently improves robustness.

Furthermore, we additionally evaluate multi-model (offline) priors obtained by aggregating confusion statistics from an ensemble of three independently trained source-domain models and freezing this prior during training. As summarized in Table 16, on [A]→[C] mIoU increases from 38.7 (no prior) to 40.7 with an offline prior and to 41.0 with an ensemble-based offline prior; on [B]→[C], mIoU rises from 24.9 to 26.4 and 26.9, respectively. Nonetheless, both variants remain inferior to our online prior—which co-evolves with the current model's decision boundary—achieving 41.3 on [A]→[C] and 27.2 on [B]→[C]. We attribute this gap to a fundamental mismatch: offline priors are static and cannot track the evolving decision boundary, calibration, and feature geometry during training (especially under strong augmentations), whereas our online prior updates on the original domain and is only frozen for detection on augmented data, preserving alignment throughout training. These findings are consistent with our analysis in the main text (lines 456–467; see also Table 6).

| Strategy | [A]→[C] | [B]→[C] |
|---|---|---|
| None | 38.7 | 24.9 |
| Offline | 40.7 | 26.4 |
| Offline (ensemble) | 41.0 | 26.9 |
| Online (ours) | **41.3** | **27.2** |

Table 16: Comparison of offline and online confusion priors. Online (ours) consistently yields the best mIoU by maintaining alignment with the evolving decision boundary.

## P    ORACLE 0.0 IOU ON RARE CLASSES VS. A3POINT IMPROVEMENTS

Revisiting the Oracle results on [C], the observed 0.0 IoU for *motorcycle/motorcyclist* arises from reproducibility instability under extreme class imbalance and cross-weather variation. Multiple independent reruns yield non-zero IoUs with substantial variance, indicating that 0.0 is a variance artifact rather than a systematic failure. Five independent runs are shown below.

| Run | car | bi.cle | mt.cle | truck | oth-v. | pers. | bi.clst | mt.clst | road | parki. | sidew. | other-g. | build. | fence | veget. | trunk | terr. | pole | traf. | mIoU |
|---|---|---|---|---|---|---|---|---|---|---|---|---|---|---|---|---|---|---|---|---|
| 1 | 89.4 | 42.1 | 0.0 | 59.9 | 61.2 | 69.6 | 39.0 | 0.0 | 82.2 | 21.5 | 58.2 | 45.6 | 86.1 | 63.6 | 80.2 | 52.0 | 77.6 | 50.1 | 61.7 | 54.7 |
| 2 | 89.6 | 37.4 | 9.6 | 56.4 | 64.3 | 67.3 | 44.8 | 0.8 | 81.8 | 27.9 | 54.6 | 50.1 | 86.5 | 66.2 | 78.3 | 56.3 | 75.4 | 45.8 | 58.6 | 55.3 |
| 3 | 89.7 | 41.8 | 4.0 | 58.6 | 62.8 | 66.1 | 40.7 | 0.0 | 82.0 | 23.4 | 51.8 | 47.2 | 86.7 | 64.9 | 78.7 | 59.1 | 75.0 | 49.0 | 57.0 | 54.6 |
| 4 | 89.6 | 35.3 | 0.2 | 60.8 | 64.3 | 67.4 | 44.4 | 5.4 | 81.3 | 27.4 | 54.6 | 50.2 | 87.2 | 66.3 | 77.8 | 61.1 | 74.0 | 51.8 | 59.5 | 55.7 |
| 5 | 89.7 | 39.9 | 6.8 | 58.1 | 62.1 | 61.6 | 48.5 | 11.1 | 76.4 | 21.7 | 51.4 | 43.5 | 83.1 | 61.3 | 77.2 | 63.4 | 74.9 | 55.3 | 56.4 | 54.9 |

Table 17: Oracle on [C]: per-class IoU across five independent runs. Rare classes (motorcycle, motorcyclist) exhibit high variance; 0.0 IoU appears in some runs but not others.

This instability also explains why other methods (including the baseline) show noticeable gains on *motorcycle/motorcyclist*: training on source domain [A] supplies stable, well-represented examples that mitigate scarcity on [C]. The effect is dataset-driven and broadly observable, not exclusive to a single algorithm.

Beyond dataset coverage, A3Point contributes two components that particularly help long-tailed, small-footprint classes under strong perturbations:

- **Exposure via EAS.** The augmentation space diversifies point density and pose, improving recall for fragile classes that otherwise suffer from sparsity and occlusion at long range.

- **Safer supervision via SSR.** When augmentations induce semantic shift, SSR replaces hard supervision with latent-space distillation toward the best-matching code. This prior-consistent guidance reduces label–observation misalignment and stabilizes training for rare categories.

In practice, augmentation-heavy baselines already improve rare classes relative to a plain Oracle; SSR's latent guidance further reduces false positives and harmful updates. The net effect is steadier

gains on *motorcycle/motorcyclist* and similar rare categories. The revised text clarifies the distinction between dataset-induced improvements and A3Point-specific stabilization.

## Q    EFFECTIVENESS FOR SAFELY ALLOWING A LARGER AUGMENTATION SPACE

We add experiments to directly assess whether A3Point enables safe expansion of the augmentation space.

- **Scope of the default EAS.** The default ranges already span common street-scene Li-DAR degradations. Pushing beyond these bounds (e.g., extreme point-drop ratios or jitter) quickly becomes physically implausible and risks label–geometry mismatch, which can corrupt supervision.
- **Stress test with arbitrarily large ranges.** We expand jitter std to $[0, 0.10]$ and point-drop ratio to $[0, 0.99]$ (none–excessive), keep all other augmentations unchanged, and evaluate on [A]→[C].

| Method | None | light | moderate | heavy | random (light–heavy) | excessive | random* (none–excessive) |
|--------|------|-------|----------|-------|----------------------|-----------|--------------------------|
| Baseline | 31.4 | 37.6 | 38.0 | 37.1 | 38.7 | 5.1 | 36.5 |
| A3Point | 31.8 | 38.5 | 40.4 | 40.5 | **41.3** | 7.8 | **41.2** |

Table 18: Stress test with enlarged augmentation ranges on [A]→[C]. Random: sampled within light–heavy; Random*: sampled within none–excessive.

As shown in Table 18, we have key observations:

- Excessive augmentation without SSR is harmful. With very large ranges, the baseline suffers severe degradation, consistent with mis-supervision induced by semantic shift.
- SSR stabilizes training under heavy distortions. A3Point maintains strong performance when sampling randomly across none→excessive, because SSR localizes shifted regions and switches to latent-space distillation to avoid harmful updates.
- There is a hard upper bound set by physics. When drop ratio $\to 1$ and jitter std $\to 0.1$, inputs cease to resemble plausible LiDAR scans; performance drops for all methods, indicating the practical limit of useful augmentation.
- **Evidence of adaptive behavior.** As shown in Fig.5, the SSR mask ratio increases with augmentation strength, indicating that the detector adaptively flags more regions as shift intensifies. This adaptivity enables safe use of stronger augmentations: regions consistent with the learned confusion prior receive standard supervision, while shifted regions receive soft, prior-consistent guidance.

In summary, A3Point expands the usable augmentation envelope by mitigating mis-supervision where semantic shift occurs, while respecting a physically grounded ceiling beyond which augmentation is no longer beneficial.

## R    EXTENSION: ITERATIVE A3POINT WITH VARYING AUGMENTATION SPACE

Iteratively refining a confusion prior (e.g., a confusion matrix or SCP) can further enhance robustness and generalization. The main limitation of purely offline priors is their static nature: similar to the multi-model ensemble in Q1, they cannot track the evolving decision boundary, calibration, and capacity of the current network. In practice, such static priors often misalign with the model's confidence distribution and error modes—especially early in training—leading to inferior results compared with online evolution.

In contrast, our approach updates the confusion prior online on the original domain and freezes it only for anomaly detection on augmented inputs. This design does not require a perfect prior;

instead, it maintains alignment with the model's current state throughout training. Empirically, the adaptive online prior yields better stability and higher final mIoU than offline confusion matrices (see Table 6), with particularly larger gains for long-tailed classes and long-range sparse regions.

To evaluate a curriculum-style extension, we implement an iterative variant of A3Point that expands the augmentation space in stages. The training proceeds in three phases: start with a light augmentation ceiling and train for several epochs; use the resulting SCP as initialization and raise the ceiling to moderate; finally, proceed to a heavy ceiling. Thus, the augmentation ceilings progress light → moderate → heavy, and each stage initializes with the SCP learned in the previous one. Results are summarized below.

| Strategy | [A]→[C] | [B]→[C] |
|---|---|---|
| None | 38.7 | 24.9 |
| A3Point | 41.3 | 27.2 |
| A3Point+ (curriculum) | **41.5** | **27.8** |

Table 19: Iterative, curriculum-style A3Point that expands the augmentation ceiling in stages (light → moderate → heavy).

As shown in Table 19, the iterative scheme offers modest additional gains when expanding to a larger augmentation space. It is a promising direction—using SCP statistics to adapt augmentation distributions and SSR thresholds in a closed-loop schedule—while the principal improvements still stem from maintaining online alignment rather than from offline aggregation.

## S    STRONGER AUGMENTATION FOR PREVIOUS METHODS

To ensure fairness, prior augmentation-based methods are re-run with the same upper bounds as our Enhanced Augmentation Space (EAS): point drop ratio sampled in $[0.2, 0.8]$ and jitter $\sigma$ in $[0.01, 0.05]$, with their original pipelines otherwise unchanged.

| Method | [A]→[C] | [B]→[C] |
|---|---|---|
| Baseline | 31.4 | 15.5 |
| Baseline + EAS | 38.7 | 24.9 |
| PointDR* | 33.9 | 19.8 |
| PointDR* + EAS | 39.1 | 25.1 |
| LiDARWeather* | 37.8 | 20.4 |
| LiDARWeather* + EAS | 39.6 | 25.9 |
| A3Point (ours) | **41.3** | **27.2** |

Table 20: Re-running prior methods under the same EAS upper bounds. * denotes the reproduced result.

Under stronger augmentations, prior methods do improve, but gains saturate and remain below A3Point. Expanding the augmentation space alone is insufficient unless semantic shift is explicitly handled; SSR localization and latent distillation make stronger augmentation genuinely usable.

## T    VISUALIZING THE LEARNED CODEBOOK

We visualize the learned VQ-VAE codebook using t-SNE, as shown in Fig.11. Each per-class sub-codebook is rendered in a distinct color; marker size encodes the per-code variance tracked via EMA, providing an at-a-glance sense of prototype stability and coverage.

Consistent patterns observed in the visualizations:

- **Intra-class multimodality.** Within each class, prototypes form multiple clusters that align with distinct local geometry and context configurations (e.g., road–sidewalk bound-

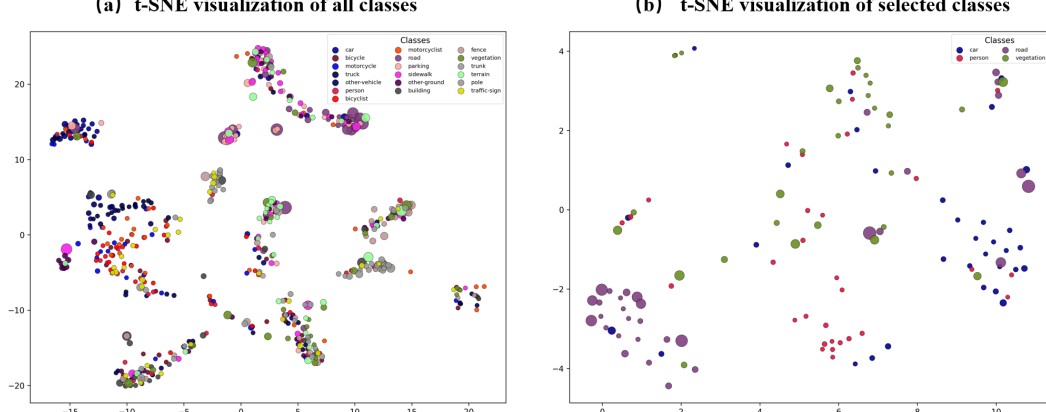

Figure 11: t-SNE of the learned VQ-VAE codebook. Colors denote class-specific sub-codebooks; marker size encodes per-code variance.

aries, car–vegetation occlusion edges, terrain discontinuities). The codebook captures fine-grained, recurring confusion modes rather than collapsing to a single class template.

- **Inter-class neighborhoods.** Prototypes from semantically similar or commonly confused categories (e.g., *road* vs. *sidewalk*, *building* vs. *fence*) lie in proximity, reflecting typical confusion manifolds encountered by the segmentation network.

- **Progressive specialization.** Over training, prototypes become more compact and better separated along confusion-specific axes, indicating increasing specialization. This sharpening improves anomaly detection for semantic shift: embeddings that deviate from well-formed confusion clusters are more easily identified as outliers.

By encoding confusion manifolds explicitly in latent space, the codebook provides a principled basis for distinguishing genuine semantic confusion (expected, class-consistent variability) from augmentation-induced semantic shift (label–geometry mismatch). Rather than memorizing class means, it organizes a discrete atlas of confusion patterns that SSR localization can leverage to decide when predictions are consistent versus shifted.

## U  DIRECT COMPARISONS WITH OTHER DG METHODS.

We conduct controlled comparisons within the same augmentation space as A3Point. Specifically:

- **Mean Teacher on augmentations (MT):** the student receives strongly augmented inputs, while the teacher sees the original or weakly augmented inputs; a KL divergence consistency loss is applied.

- **Symmetric Cross Entropy (SCE):** a label-noise robust objective.

| Method | [A]→[C] | [B]→[C] |
|---|---|---|
| Baseline | 31.4 | 15.5 |
| EAS | 38.7 | 24.9 |
| EAS + MT | 39.0 | 25.5 |
| EAS + SCE | 38.8 | 24.6 |
| A3Point (ours) | **41.3** | **27.2** |

Table 21: Comparisons with other DG methods under the same augmentation space.

As shown in Table 21, we have observations:

- MT and SCE yield modest gains but do not explicitly identify or handle augmentation-induced semantic shift under heavy perturbations, leading to overfitting to misaligned pseudo-targets or necessitating weaker augmentations.

- A3Point combines (i) class-wise semantic confusion priors learned from unaugmented predictions, (ii) latent-space anomaly detection to localize semantic-shift regions (SSR), and (iii) region-adaptive supervision, yielding more stable improvements and higher mIoU on heavy-weather targets.

## V    WHY GLOBAL NEAREST CODE FOR SSR?

**Motivation.**   Heavy augmentations can introduce label–semantics misalignment during SSR. Using class-conditional nearest code anchors supervision to the original class codebook, potentially reinforcing bias from incorrect labels. In contrast, a *global* nearest code provides a class-agnostic, stability-oriented prior that aligns shifted regions to the most compatible confusion mode in latent space, without being dragged by the (possibly wrong) original class.

| Variant | [A]→[C] | [B]→[C] |
|---|---|---|
| Baseline | 31.4 | 15.5 |
| EAS | 38.7 | 24.9 |
| Class-conditional nearest code (SSR) | 38.8 | 24.5 |
| Soft CE on logits with temperature (SSR) | 39.7 | 25.8 |
| Consistency with pre-augmentation prediction | 37.2 | 24.4 |
| A3Point (global nearest code in latent space) | **41.3** | **27.2** |

Table 22: Controlled comparisons of SSR supervision choices under the same augmentation space. Global nearest code (class-agnostic latent alignment) performs best.

**Interpretation.**   We perform controlled comparisons in Table 22:

- **Class-conditional nearest code.** Keeps supervision tied to the original class-specific codebook. It does not resolve label–semantics mismatch under strong augmentations and remains susceptible to original-class bias, yielding negligible improvement over EAS.

- **Soft CE with temperature on logits.** Temperature-scaled soft cross-entropy reduces overconfidence and partially alleviates semantic shift, giving slight gains. However, it remains label-conditioned and cannot fully correct mismatches.

- **Pre-augmentation consistency.** Using the pre-augmented prediction as a teacher is vulnerable when augmentations induce semantic shift; enforcing such consistency reinforces an unreliable alignment target, degrading performance relative to EAS.

**Conclusion.**   Our class-agnostic objective—distillation toward the *global* nearest code in latent space for SSR—explicitly decouples supervision from potentially incorrect labels and aligns shifted regions with the most compatible semantic confusion prior, consistently achieving the best performance.

## W    CLASS MAPPING ACROSS DATASETS

We follow the standard 19-class evaluation protocol shared by SemanticKITTI (source [A]), Syn-LiDAR (source [B]), and SemanticSTF (target [C]). All datasets are mapped to a unified 19-class taxonomy with IDs 0–18; ID 255 denotes "ignore" and is excluded from loss and metrics.

### W.1    UNIFIED 19-CLASS TAXONOMY

```
ID → Class:
0 car, 1 bicycle, 2 motorcycle, 3 truck, 4 other-vehicle,
5 person, 6 bicyclist, 7 motorcyclist, 8 road, 9 parking,
```

```
10 sidewalk, 11 other-ground, 12 building, 13 fence,
14 vegetation, 15 trunk, 16 terrain, 17 pole, 18 traffic-sign
```

## W.2 [A] SEMANTICKITTI: NATIVE LABELS AND KEPT SET

**Label-to-name (native).**

```
label_name_mapping = {
0: 'unlabeled',
1: 'outlier',
10: 'car',
11: 'bicycle',
13: 'bus',
15: 'motorcycle',
16: 'on-rails',
18: 'truck',
20: 'other-vehicle',
30: 'person',
31: 'bicyclist',
32: 'motorcyclist',
40: 'road',
44: 'parking',
48: 'sidewalk',
49: 'other-ground',
50: 'building',
51: 'fence',
52: 'other-structure',
60: 'lane-marking',
70: 'vegetation',
71: 'trunk',
72: 'terrain',
80: 'pole',
81: 'traffic-sign',
99: 'other-object',
252: 'moving-car',
253: 'moving-bicyclist',
254: 'moving-person',
255: 'moving-motorcyclist',
256: 'moving-on-rails',
257: 'moving-bus',
258: 'moving-truck',
259: 'moving-other-vehicle'
}
```

**Kept labels for the 19-class protocol.**

```
kept_labels = [
'road','sidewalk','parking','other-ground','building','car','truck',
'bicycle','motorcycle','other-vehicle','vegetation','trunk','terrain',
'person','bicyclist','motorcyclist','fence','pole','traffic-sign'
]
```

**Notes.** Moving classes (e.g., 252–259) are mapped to their static counterparts (e.g., moving-car → car) before applying the 19-class filter, following the official protocol. Categories outside the 19-class set are assigned to 255 (ignore).

## W.3 [B] SYNLIDAR → UNIFIED 19-CLASS IDs

```
learning_map:
0: 255 # unlabeled → ignore
1: 0 # car → car
2: 3 # pick-up → truck
3: 3 # truck → truck
```

```
4: 4 # bus → other-vehicle
5: 1 # bicycle → bicycle
6: 2 # motorcycle → motorcycle
7: 4 # other-vehicle → other-vehicle
8: 8 # road → road
9: 10 # sidewalk → sidewalk
10: 9 # parking → parking
11: 11 # other-ground → other-ground
12: 5 # female → person
13: 5 # male → person
14: 5 # kid → person
15: 5 # group → person
16: 6 # bicyclist → bicyclist
17: 7 # motorcyclist → motorcyclist
18: 12 # building → building
19: 255 # other-structure → ignore
20: 14 # vegetation → vegetation
21: 15 # trunk → trunk
22: 16 # terrain → terrain
23: 18 # traffic-sign → traffic-sign
24: 17 # pole → pole
25: 255 # traffic-cone → ignore (not in 19-class)
26: 13 # fence → fence
27: 255 # garbage-can → ignore
28: 255 # electric-box → ignore
29: 255 # table → ignore
30: 255 # chair → ignore
31: 255 # bench → ignore
32: 255 # other-object → ignore
```

## W.4 [C] SEMANTICSTF → UNIFIED 19-CLASS IDs

```
learning_map = {
0: 255, # unlabeled → ignore
1: 0, # car
2: 1, # bicycle
3: 2, # motorcycle
4: 3, # truck
5: 4, # other-vehicle
6: 5, # person
7: 6, # bicyclist
8: 7, # motorcyclist
9: 8, # road
10: 9, # parking
11: 10, # sidewalk
12: 11, # other-ground
13: 12, # building
14: 13, # fence
15: 14, # vegetation
16: 15, # trunk
17: 16, # terrain
18: 17, # pole
19: 18, # traffic-sign
20: 255 # invalid → ignore
}
```

## X LLMS AND SOCIETY IMPACT

Use of Large Language Models (LLMs). In preparing this manuscript, we used LLMs solely for language polishing and writing assistance (e.g., clarity, grammar, and style). LLMs were not used to generate research ideas, experimental results, code, or analyses, and no proprietary data or sensitive information were provided to LLMs beyond the manuscript text. All technical content, experiments, and conclusions were produced and verified by the authors.

Within this paper, we present an approach for domain-generalized LiDAR semantic segmentation under adverse weather. Our contributions focus on robustness to distribution shifts via augmentation-aware latent learning and semantic shift localization. At present, we are not aware of direct negative societal implications arising specifically from the proposed methodology.

