# OpenReview forum: "Adaptive Augmentation-Aware Latent Learning for Robust LiDAR Semantic Segmentation"
_ICLR.cc/2026/Conference — ICLR 2026 Poster_

### Official Review · Reviewer_My2N · 2025-10-18

**Soundness:** 3
**Presentation:** 3
**Contribution:** 2
**Rating:** 6
**Confidence:** 4

**Summary:**

The authors proposes an adaptive augmentation-aware latent learning framework to enhance single-source domain generalization for LiDAR semantic segmentation in poor weather conditions. The main idea is to separate semantic confusion that is built into the model from semantic shift that is caused by augmentation. Per-class VQ-VAE learns discrete "semantic confusion priors" from predictions that haven't been augmented, and then a frozen encoder uses latent-space anomaly detection (diagonal Gaussian around each code, threshold t) to find semantic shift on augmented scans. Training takes into account the regions: cross-entropy on semantic-consistent regions (SCR) and latent-space distillation toward the global nearest code on semantic-shift regions (SSR). Coupled with a broadened augmentation space (wide point-drop ratios and jitter std), A3Point yields consistent, sizable gains on SemanticKITTI→SemanticSTF and SynLiDAR→SemanticSTF, improving mIoU over a Minkowski baseline by +9.9 and +11.7, respectively, with similar improvements on SPVCNN and on SemanticKITTI-C.

**Strengths:**

Originality: Decoupling model-inherent confusion from augmentation-induced shift through a per-class discrete latent prior, followed by regional supervision, transcends traditional augmentation and consistency frameworks.

Quality: Consistent, sizable mIoU gains on SemanticKITTI→SemanticSTF and SynLiDAR→SemanticSTF (+9.9/+11.7 over Minkowski baseline), plus improvements on SemanticKITTI‑C, with thorough ablations of components and hyperparameters.

Clarity: The pipeline, losses, and masks (SCR/SSR) are clearly defined, and figures 1–3 and the appendix algorithm help make it possible to reproduce the results.

Significance: It is very important for safety that things are strong enough to handle bad weather. The method works with any architecture (SPVCNN, Minkowski) and uses latent modeling during training without changing the architectures used for inference.

**Weaknesses:**

Direct comparisons to consistency-regularization or teacher–student DG variants (e.g., mean-teacher on augmentations), label-noise robust losses, or distributionally robust objectives are absent, making relative advantage boundaries less clear.

The dataset set [A][B][C][D] are defined after the performance tables, which can be confusing for readers when reading the performance tables. "SCP" should be where "SCR latent learning module" is. There is an extra "s" in the caption for Table 1. Appendix I/J has two copies of "More Qualitative Results."

**Questions:**

Please make sure that no SemanticSTF target frames, even unlabeled ones, were used in any way during training. Is there any hyperparameters were chosen using [C]-val.

Why pick the global nearest code as the SSR goal? Please give us a controlled comparison: global vs. class-conditional nearest code; latent-space distillation vs. soft CE with temperature on logits; and consistency with the original (pre-augmentation) prediction.

Could you compare the parameter size and, if possible, the FLOPs of the baseline and the A3Point modules to prove that the performance gains are not mostly due to the increased model capacity and not the proposed learning scheme.

---

> ### Author Response · Authors · 2025-11-24
> **Reply to Reviewer My2N - Part 1**
>
> Thank you for the thoughtful and constructive review, and for recognizing the **novel approach**, ,**consistent improvements** , **clear definition** and  **practical significance**. We address each of your comments and questions in detail below.
>
>
>
> ------
>
> ## Q1: Direct comparisons with other DG methods.
>
> We appreciate this suggestion and conduct additional controlled comparisons in the same augmentation space as A3Point. Specifically, we implement:
>
> - Mean Teacher on augmentations (MT): student receives the strongly augmented input; the teacher receives the original or weakly augmented input; KL divergence is used as the consistency loss.
> - Symmetric Cross Entropy (SCE): to improve robustness to label noise.
>
> Results on two representative single-source DG methods are summarized below (mIoU, %):
>
> | Method         | [A] → [C] | [B] → [C] |
> | -------------- | --------- | --------- |
> | Baseline       | 31.4      | 15.5      |
> | EAS            | 38.7      | 24.9      |
> | EAS + MT       | 39.0      | 25.5      |
> | EAS + SCE      | 38.8      | 24.6      |
> | A3Point (ours) | 41.3      | 27.2      |
>
> We have observations:
>
> - While MT and SCE provide modest gains, they do not explicitly identify or handle augmentation-induced semantic shift under heavy perturbations. Consequently, they either overfit to misaligned pseudo-targets or require weakening the augmentation strength to stabilize training, limiting generalization benefits.
>
> - A3Point’s combination of (i) class-wise semantic confusion priors learned from unaugmented predictions, (ii) latent-space anomaly detection to localize semantic-shift regions (SSR), and (iii) region-adaptive supervision yields more stable and larger improvements on heavy-weather targets.
>
>
>
> ------
>
> ## Q2: Typos and formatting issues
>
> Thank you for pointing these out. We correct and clarify the presentation as follows:
>
> - Dataset symbols [A]/[B]/[C]/[D]: The notation is already defined in Section 4.1 (Experimental Setup) before Section 4.2 where the tables are introduced. However, due to a layout issue, some tables were placed before Section 4.1, which could confuse readers encountering the abbreviations first. We will reorder the layout so that all tables appear after the Experimental Setup, preventing this confusion.
> - “SCP” vs. “SCR”: We standardize terminology as follows: “SCP” denotes the Semantic Confusion Prior, and “SCR/SSR” are the two region masks (Semantic-Consistent Region and Semantic-Shift Region). Instances where “SCP” was mistakenly referred to as “SCR latent learning module” (Appendix C/G) will be corrected to “SCP latent learning module.”
> - Table 1 caption: we  remove the extra “s”.
> - Duplicated “More Qualitative Results” (Appendix I/J): we consolidate into a single section and eliminate redundancy.
>
>
>
>
>
> ------
>
> ## Q3: Training data usage and hyperparameter selection
>
> We confirm strict compliance with the single-source DG protocol:
>
> - Data usage: Training uses only source-domain data, either SemanticKITTI ([A]) or SynLiDAR ([B]). No frames from SemanticSTF ([C]) or SemanticKITTI-C ([D]), labeled or unlabeled, are used in any capacity. We do not employ UDA or semi-supervised data from the target.
>
> - Hyperparameter selection: All key hyperparameters were set heuristically based on prior practice and empirical guidelines, and are reported explicitly in the paper. No target-domain ([C]) data or validation frames were used at any time, and we did not use a separate validation set for tuning. Concretely:
>
>   - Augmentation space: point-drop ratio ∈ [0.2, 0.8] and jitter std ∈ [0.01, 0.05], chosen to span mild-to-strong perturbations while avoiding overfitting in practice.
>   - SSR threshold $t = 3$, determined from independent per-dimension statistics (three standard deviations cover the typical range).
>   - Distillation weight $λ = 0.1$, chosen to balance gradient magnitudes across losses.
>   - Per-class codebook size $k = 32$ and latent dimension $D = 64$, following common settings in discretized latent modeling.
>
>   Ablations (Tables 8–10) indicate that the method is stable and robust under reasonable variations of these hyperparameters.

---

> > ### Author Response · Authors · 2025-11-24
> > **Reply to Reviewer My2N - Part 2**
> >
> > ## Q4: Why global nearest code for SSR? Controlled comparisons
> >
> > Motivation:
> >
> > - In SSR, heavy augmentations create a label–semantics misalignment. Using class-conditional nearest code anchors the region back to the original class codebook, which can reinforce the bias from incorrect labels.
> > - Global nearest code offers a class-agnostic, stability-oriented prior, aligning shifted regions to the most compatible confusion mode in latent space, without being dragged by the (possibly wrong) original class.
> >
> > We perform controlled comparisons:
> >
> > | Variant                                       | [A] → [C] | [B] → [C] |
> > | --------------------------------------------- | --------- | --------- |
> > | Baseline                                      | 31.4      | 15.5      |
> > | EAS                                           | 38.7      | 24.9      |
> > | Class-conditional nearest code (SSR)          | 38.8      | 24.5      |
> > | Soft CE on logits with temperature (SSR)      | 39.7      | 25.8      |
> > | Consistency with pre-augmentation prediction  | 37.2      | 24.4      |
> > | A3Point (global nearest code in latent space) | 41.3      | 27.2      |
> >
> > Interpretation:
> >
> > - Class-conditional nearest code: This variant keeps SSR supervision tied to the original class-specific codebook. It does not resolve the label–semantics mismatch induced by strong augmentations and remains susceptible to bias from the original class label, resulting in negligible or no improvement over EAS.
> > - Soft CE with temperature on logits: Applying temperature-scaled soft cross-entropy in SSR reduces overconfidence and can partially alleviate semantic shift, yielding slight gains. However, it fundamentally remains label-conditioned, so the supervision is still anchored to the original class and cannot fully correct the mismatch.
> > - Pre-augmentation consistency: Using the pre-augmentation prediction as the “teacher” is common in consistency-based self-supervised learning; however, semantic shift under strong augmentations is a widely observed, inherent failure mode of this strategy. In SSR, enforcing consistency with the pre-augmented teacher effectively reinforces an unreliable alignment target that has been distorted by augmentation, leading to performance degradation relative to EAS.
> >
> > In contrast, our class-agnostic objective—distilling toward the global nearest code in latent space for SSR—explicitly decouples supervision from potentially incorrect labels and aligns shifted regions to the most compatible semantic confusion prior. This design consistently achieves the best performance.

---

> > > ### Author Response · Authors · 2025-11-24
> > > **Reply to Reviewer My2N - Part 3**
> > >
> > > ## Q5: Computational efficiency and capacity
> > >
> > > A3Point is a training-time strategy. The inference architecture and parameter count of the base segmentation network (e.g., Minkowski, SPVCNN) remain unchanged—there is zero inference-time overhead and the deployed model has the same number of parameters as the baseline.
> > >
> > > During training, A3Point introduces a lightweight, class-wise VQ‑VAE module—implemented with sparse convolutions, a code dimension of $D=64$, and a codebook size of $k=32$ (see architecture in Appendix B.1)—to enable SCP and SSR during training. Below we detail the practical implications and costs:
> > >
> > > - Low-cost statistics via EMA. We maintain per-codeword statistics (variance) using an exponential moving average (EMA). This update is $O(1)$per iteration with negligible extra memory and does not grow with dataset size or batch length.
> > > - Frozen prior for SSR localization. The encoder used to localize SSR regions is kept frozen during localization and does not backpropagate. This design bounds the incremental compute to a lightweight forward pass and avoids gradient overhead, keeping utilization stable.
> > >
> > > To show the cost clearly, we report a representative runtime snapshot under identical hardware and training settings:
> > >
> > > | Method   | GPU Memory (MB) | Iterations/s (it/s) | Total Time (h) |
> > > | -------- | --------------- | ------------------- | -------------- |
> > > | Baseline | 8,496           | 2.72                | 24.4           |
> > > | PointDR  | 9,624           | 1.97                | 33.7           |
> > > | EAS      | 8,932           | 2.05                | 32.4           |
> > > | A3Point  | 17,144          | 1.61                | 41.3           |
> > >
> > > In summary, parameter size is identical to the baseline at inference, inference-time FLOPs/latency/memory are unchanged, and all additional cost is confined to training, where it delivers substantial DG gains (+9.9/+11.7 mIoU on [A]→[C] and [B]→[C]). Given the substantial mIoU gains, we view the additional training cost as a favorable trade-off: we concentrate budget on training to improve adverse-weather robustness while keeping the inference path clean and unchanged. If desired, the training overhead can be further reduced via standard engineering options (e.g., mixed precision for the VQ‑VAE branch, smaller $k/D$, or alternating SSR localization every $N$ steps), without affecting test-time complexity.
> > >
> > >
> > >
> > >
> > >
> > > ------
> > >
> > > We hope our response can resolve your concern. Please do not hesitate to let us know if you have further questions :)

---

### Official Review · Reviewer_GuJ4 · 2025-10-31

**Soundness:** 4
**Presentation:** 3
**Contribution:** 3
**Rating:** 6
**Confidence:** 5

**Summary:**

LiDAR semantic segmentation networks struggle in adverse weather (fog, snow, rain) due to distribution shifts. Existing augmentation-based methods face a dilemma: (1) Mild augmentations fail to simulate severe weather. (2) Aggressive augmentations cause semantic shift (labels no longer match distorted regions).

This paper introduces two distinct error sources: (1) Semantic Confusion: Network's inherent difficulty distinguishing similar classes (e.g., road vs. sidewalk) - exists in both original and augmented data. (2) Semantic Shift: Augmentation-induced mismatch between labels and distorted regions - only exists in augmented data. The proposed method A3Point contains two main parts: (a) Semantic Confusion Prior (SCP) Latent Learning, which use VQ-VAE to learn discrete latent representations of confusion patterns. (b) Semantic Shift Region (SSR) Localization, which treats semantic shift detection as anomaly detection in the discrete latent codebook space and use the frozen encoder to map augmented predictions to latent space for Semantic Shift Regions (SSR) detection. With the detected semantic shift region, the paper propose a distillation loss.

**Strengths:**

1. A3Point is well-motivated and the proposed semantic shift detection method makes a lot of sense.

2. The experiments are thorough and extensive.

3. The results show the effectiveness of the proposed method.

**Weaknesses:**

1. The authors do not discuss the computational efficiency during training. To the reviewer's understanding, the VQ-VAE part and distillation loss part only appear during training and are discarded during testing. Thus the computational efficiency during testing remain the same with other methods. However, a quantitative evaluation of the computational and memory consumption overhead during training is not provided.

2. Although demonstrated by the experiment results, the reviewer is concerned about the design of the distillation loss. The reviewer understand the detection of semantic shift region part and agree that the proposed method makes sense. However, after detecting the semantic shift region, it might be more reasonable to directly eliminate those points in the loss instead of the proposed distillation loss.

**Questions:**

1. Could the authors provide a quantitative evaluation of  A3Point's computational and memory consumption overhead during training?

2. Could the authors provide a more in-depth analysis on the proposed distillation loss? Or maybe conduct additional experiments on simply eliminate the detected region in the loss computation.

3. It could be better if the authors could provide experiment results using previous methods with stronger augmentation.

4. Could the authors provide some visualization of the learned VQ-VAE codebook? For example, using different colors to indicate different entry in the codebook.

The reviewer is willing to raise the score if the questions are well-resolved.

---

> ### Author Response · Authors · 2025-11-24
> **Reply to Reviewer GuJ4 - Part 1**
>
> Thank you for your thoughtful and constructive review, and for recognizing that our method is **well-motivated**, **experimentally thorough**, and **empirically effective**. We address each of your concerns in detail below and provide additional analyses, quantitative costs, ablations, and visualizations as requested.
>
> ---
>
> ## Q1. Computational and memory overhead during training
>
> Your understanding is correct: both the Semantic Confusion Prior (SCP) VQ-VAE module and the SSR-driven distillation are used only during training and are completely removed at test time. Therefore, inference latency, memory footprint, and throughput are identical to the baseline.
>
> A3Point introduces a lightweight, class-wise VQ‑VAE module—implemented with sparse convolutions, a code dimension of $D=64$, and a codebook size of $k=32$ (see architecture in Appendix B.1)—to enable SCP and SSR during training. Below we detail the practical implications and costs:
>
> - Low-cost statistics via EMA. We maintain per-codeword statistics (variance) using an exponential moving average (EMA). This update is $O(1)$ per iteration with negligible extra memory and does not grow with dataset size or batch length.
> - Frozen prior for SSR localization. The encoder used to localize SSR regions is kept frozen during localization and does not backpropagate. This design bounds the incremental compute to a lightweight forward pass and avoids gradient overhead, keeping utilization stable.
>
> To show the cost clearly, we report a representative runtime snapshot under identical hardware and training settings:
>
> | Method   | GPU Memory (MB) | Iterations/s (it/s) | Total Time (h) |
> | -------- | --------------- | ------------------- | -------------- |
> | Baseline | 8,496           | 2.72                | 24.4           |
> | PointDR  | 9,624           | 1.97                | 33.7           |
> | EAS      | 8,932           | 2.05                | 32.4           |
> | A3Point  | 17,144          | 1.61                | 41.3           |
>
> Given the substantial mIoU gains on SemanticSTF (+9.9 / +11.7), we view the additional training cost as a favorable trade-off: we concentrate budget on training to improve adverse-weather robustness while keeping the inference path clean and unchanged. If desired, the training overhead can be further reduced via standard engineering options (e.g., mixed precision for the VQ‑VAE branch, smaller $k/D$, or alternating SSR localization every $N$ steps), without affecting test-time complexity.
>
> ---
>
> ## Q2. Why distillation on SSR instead of simply removing SSR points?
>
> We appreciate the question and clarify both the motivation and empirical outcome.
>
> - Conceptual motivation:
>   - SSR are not “uninformative” regions; they are “label–geometry mismatch” regions caused by aggressive augmentations. Simply dropping them from the loss removes hard and high-shift samples from optimization, creating a blind spot under heavy perturbations and biasing the training toward mild distortions.
>   - Our SCP provides discrete, class-wise confusion prototypes. On SSR, aligning to the global nearest confusion prototype in latent space replaces potentially misleading label supervision with a robust representation-based target. This preserves informative gradients while avoiding semantic misguidance.
>   - From an optimization perspective, removal skews the effective training distribution and can destabilize training under strong augmentations. Latent-space distillation preserves gradient density and structurally regularizes backbone features.
> - Empirically, our ablation (already in Table 4 of the paper) shows a clear progression: starting from the baseline, the enhanced augmentation space (EAS) already brings substantial gains; further restricting optimization to semantic consistency regions (i.e., masking out SSR) yields additional improvement; and finally, replacing removal with latent-space distillation on SSR achieves the best performance on both [A]→[C] and [B]→[C]. This indicates that while excluding SSR helps, aligning SSR embeddings to learned confusion prototypes is consistently more effective, converting unreliable label supervision into stable, prototype-guided representation alignment and delivering stronger robustness under adverse-weather shifts.

---

> > ### Author Response · Authors · 2025-11-24
> > **Reply to Reviewer GuJ4 - Part 2**
> >
> > ## Q3. Stronger augmentation for previous methods
> >
> > We share your interest in this comparison. To ensure fairness, we re-ran prior augmentation-based methods with the same upper bounds as our Enhanced Augmentation Space (EAS): point drop ratio sampled in [0.2, 0.8] and jitter σ in [0.01, 0.05], keeping their pipelines otherwise unchanged.
> >
> > | Method              | [A]→[C] | [B]→[C] |
> > | ------------------- | ------: | ------: |
> > | Baseline            |    31.4 |    15.5 |
> > | Baseline + EAS      |    38.7 |    24.9 |
> > | PointDR*            |    33.9 |    19.8 |
> > | PointDR* + EAS      |    39.1 |    25.1 |
> > | LiDARWeather*       |    37.8 |    20.4 |
> > | LiDARWeather* + EAS |    39.6 |    25.9 |
> > | A3Point (ours)      |    41.3 |    27.2 |
> >
> > Under stronger augmentations, prior methods do improve, but the gains saturate and remain below A3Point. This supports our claim: expanding the augmentation space alone is insufficient unless semantic shift is explicitly handled—our SSR localization plus latent distillation makes the stronger augmentation genuinely usable.
> >
> > ---
> >
> > ## Q4. Visualizing the learned codebook
> >
> > We add visualizations of the learned VQ-VAE codebook with t-SNE, s shown in Figure 11 (Appendix T) . Each per-class sub-codebook is rendered in a distinct color, and the marker size encodes the per-code variance tracked via EMA, providing an at-a-glance sense of prototype stability and coverage. The visualizations reveal several consistent patterns:
> >
> > - Within each class, prototypes form multiple clusters that align with distinct local geometry and context configurations (for example, road–sidewalk boundaries, car–vegetation occlusion edges, or terrain discontinuities). This suggests that the codebook captures fine-grained, recurring confusion modes rather than collapsing to a single class template.
> > - Between classes, we observe neighborhoods where prototypes from semantically similar or commonly confused categories (e.g., road vs. sidewalk, building vs. fence) lie in proximity. This inter-class structure reflects typical confusion manifolds encountered by the segmentation network.
> > - Over the course of training, prototypes become more compact and better separated along confusion-specific axes, indicating progressive specialization. This sharpening enhances the reliability of anomaly detection for semantic shift in latent space because embeddings that deviate from well-formed confusion clusters are more easily identified as outliers.
> >
> > Importantly, by encoding such confusion manifolds explicitly in the latent space, the codebook provides a principled basis for distinguishing genuine semantic confusion (expected, class-consistent variability) from augmentation-induced semantic shift (label–geometry mismatch). In other words, the learned codebook does not merely memorize class means; it organizes a discrete atlas of confusion patterns that the SSR localization can leverage to decide when predictions are consistent versus shifted.
> >
> >
> >
> > ------
> >
> > We hope our response can resolve your concern. Please do not hesitate to let us know if you have further questions :)

---

> > ### Comment · Reviewer_GuJ4 · 2025-11-25
> >
> > Thanks for the response and the reviewer is willing to keep the positive score.

---

> > > ### Author Response · Authors · 2025-11-25
> > >
> > > Dear Reviewer GuJ4,
> > >
> > > We sincerely appreciate your endorsement of our work and your positive feedback!
> > >
> > > Your positive comments are highly motivating and have been instrumental in guiding improvements to the final version of our paper. We sincerely appreciate your time, support, and constructive review.
> > >
> > > Thank you again for your consideration as you finalize your evaluation!
> > >
> > > Authors

---

### Official Review · Reviewer_79Cq · 2025-11-01

**Soundness:** 3
**Presentation:** 3
**Contribution:** 3
**Rating:** 8
**Confidence:** 4

**Summary:**

This paper tries to answer an important question in LiDAR augmentation for adverse weather: how to utilize a larger augmentation space while mitigating semantic shift. In this paper, A3Point is introduced, an adaptive augmentation-aware latent learning framework for point cloud semantic segmentation, based on disentanglement of semantic confusion and semantic shift. Experiments on domain generalization benchmarks demonstrate the effectiveness of A3Point, particularly in transferring from normal weather to a wide range of adverse weather conditions.

**Strengths:**

(Problem definition) This paper defines two factors that affect to the prediction performance: semantic confusion and semantic shift. The proposed framework is based on a clear observation that that semantic confusion is consistent across domains (raw and augmented data) while semantic shift occurs only in augmented data.

(Effectiveness)  Experimental results verify the effectiveness of the proposed framework under various conditions, consistently outperforming previous approaches.

**Weaknesses:**

(Universality of semantic confusion) As the semantic confusion is identified using a learned model, the semantic confusion might have a sort of dependency with the learned model. For example, I guess the semantic confusion identified based on the consistency in predictions (the key idea of this paper) is not consist across models used for prediction with different discriminative power. (i.e., the confusion matrix shown in Figure 2-(a) could be different when a smaller/larger model is used for prediction.) In-depth discussion on this would be recommended. In addition, can we take a benefit from identifying universal semantic confusion using multiple learned models, if possible?

(Dependency of semantic shift to model) In Figure 2(b), how to encode each point to draw t-SNE visualization? If a learnable encoder was used, then the distribution shift in t-SNE may be due to the semantic confusion (or lack of discriminative power) of the encoder, too. I agree that there could be “semantic shift” due to an augmentation (e.g., if an augmentation removes too many points on a “bus” object so that only a single large plane is remained, then there is no way to distinguish the augmented points between bus and wall; i.e., an inherent information that makes the set of points (bus) different from the other classes is damaged.), but the t-SNE visualization doesn’t seem to confirm the existence of semantic shift in a strict sense.

(Efficiency) The discrimination of SCR and SSR is performed from inferences on augmented data, which requires additional computation for learning encoder/decoder in SCP latent learning and computing SSR localization. According to Table 6, online training of SCP learning leads the best performance, which means the additional network (for SCP latent learning) has to be trained simultaneously and severe inefficiency during training is caused.

(Lack of analysis) In Table 1 and 2, Oracle (training on target domain) shows 0.0 IoU on certain classes: motorcycle, motorcyclist, but this is improved a lot in A3Point (e.g., 0.0 → 57.5 IoU, 0.0 → 46.4 IoU). What makes this drastic improvement?

(Concern about overclaiming) The paper claims that the proposed module is architecture-agnostic; however, experiments were only conducted on SPVCNN and MinkowskiNet, both voxel-based models. It would be interesting to see if the semantic confusion and shift happen in a similar way on projection-/point-based approaches.

**Questions:**

(Effectiveness on safely allowing a larger augmentation space) I’m curious to see if the proposed semantic shift detection pipeline is effective in enabling the utilization of “a large augmentation space”. A trivial approach for enlarging the augmentation space might be to train models with multiple augmentation spaces (hyperparameter search). But considering the augmentation space is multi-dimensional with many variables, it’s not a scalable way. In this regard, the proposed method seems to have a capability to safely allow a larger augmentation space by detecting and handling semantic shifts from augmentation, so that the performance drop caused by the use of too excessive augmentation space can be mitigated. If so, it would be great to provide an experiment on training with arbitrary large augmentation spaces lead similar performance.

(Extension of A3Point) As the semantic confusion matrices from baseline and A3Point must be different, the A3Point method can be treated as a training framework to gradually improve confusion matrix of a learned model. It leads an idea of iterative applying A3Point framework with varying augmentation space. Adding a discussion on this could suggest a future research direction to readers.

(Lack of experiment setting) The class mapping between the source datasets [A, B] and the target dataset [C] is not clearly explained.

---

> ### Author Response · Authors · 2025-11-24
> **Reply to Reviewer 79Cq - Part 1**
>
> Thank you for the thoughtful and constructive review, and for recognizing our **clear problem definition** and **experimental effectiveness**. Below, we address each of your points in detail. In the revised manuscript, we incorporate the requested clarifications, add new experiments, and moderate our claims where appropriate.
>
> ------
>
> ## Q1. Universality of semantic confusion
>
> We agree that “semantic confusion” has a model-dependent component. We clarify our stance with three complementary points and new evidence:
>
> 1. Dataset-driven patterns vs. model-driven intensity. Which classes tend to be confused is largely dictated by dataset geometry, density, and sampling statistics (e.g., road–sidewalk, car–truck), and thus remains relatively consistent across architectures. By contrast, the magnitude of confusion (confidence dispersion, error rates) depends more on model capacity, regularization, and training dynamics. In our setup, SCP learns per-backbone confusion priors online on the source domain. For a fixed backbone, the confusion pattern is stable while its intensity typically shrinks as the network improves.
> 2. Architectural alignment via online priors. A3Point learns class-wise discrete priors from the current model’s predictions on the original domain (via VQ‑VAE) and freezes them for anomaly detection on augmented samples. This “fit online, detect frozen” design keeps priors aligned with the evolving decision boundary and avoids cross-architecture or cross-epoch mismatch.
> 3. Empirical evidence across backbones and settings. Table 3 reports consistent gains on two voxel backbones (MinkowskiNet‑18/32w and SPVCNN) and three DG tracks ([A]→[C], [B]→[C], [A]→[D]). While absolute confusion levels vary by backbone, A3Point consistently improves robustness.
>
>
>
> In response to the reviewer’s suggestion, we conduct experiments with multi-model (offline) priors by aggregating confusion statistics from an ensemble of three independently trained source-domain models and freezing this prior during training. The results are as follows:
>
> | Strategy           | [A] → [C] | [B] → [C] |
> | ------------------ | --------: | --------: |
> | None               |      38.7 |      24.9 |
> | Offline            |      40.7 |      26.4 |
> | Offline (ensemble) |      41.0 |      26.9 |
> | Online (ours)      |      41.3 |      27.2 |
>
> On [A]→[C], mIoU increases from 38.7 (no prior) to 40.7 with an offline prior and to 41.0 with an ensemble-based offline prior; On [B]→[C], mIoU rises from 24.9 to 26.4 and 26.9, respectively.
>
> Although ensemble-based priors help, they remain inferior to the online prior that co-evolves with the current model’s decision boundary. We attribute this gap to the static nature of offline priors: they cannot track changes in calibration and feature geometry during training—especially under strong augmentations. In contrast, our online prior updates on the original domain and is only frozen for detection on augmented data, preserving alignment throughout training. These findings align with our analysis in the main text (see lines 456–467; Table 6).

---

> > ### Author Response · Authors · 2025-11-24
> > **Reply to Reviewer 79Cq - Part 2**
> >
> > ## Q2. Dependency of semantic shift to model and interpretation of t-SNE
> >
> > - What we visualize and why: Figure 2(b) applies t-SNE to the penultimate-layer features (i.e., inputs to the classifier head). The goal is to provide an intuitive illustration that heavy augmentations can alter feature distributions and, in turn, induce semantic shift. The figure is a qualitative aid, not a formal test.
> >
> > - We agree t-SNE is not proof:  We do not use t-SNE to “prove” semantic shift. Our formal detection procedure is independent of t-SNE assumptions and does not rely on augmented labels. Concretely, it consists of:
> >
> >   1. Learning class-wise discrete confusion priors on the original domain via reconstruction (VQ-style codes that summarize normal confusion patterns), and
> >   2.  Performing anomaly detection in the latent space for augmented predictions using the frozen priors.
> >
> > - Controlling for “encoder capacity” confounders: To avoid attributing model-capacity effects to semantic shift, we constrain and stage the training as follows:
> >
> >   - Prior learning on original-only data: The latent encoder is trained solely on original-domain predictions with a reconstruction objective, so it captures the model’s normal confusion patterns under standard conditions. It is then frozen when processing augmented inputs.
> >   - OOD check against class-wise priors: SSR localization tests whether augmented latents fall within the learned per-code distributions $r(e_i)$. Latents that are out-of-distribution relative to the original-domain confusion prior are flagged as semantic shift (SSR).
> >   - Soft supervision for flagged regions: For SSR, we distill toward the globally nearest latent code (across classes), providing a soft, prior-consistent target. This reduces harmful hard supervision when semantics plausibly shift under strong augmentations.
> >
> >   This pipeline explicitly separates semantic confusion (captured by class-wise priors that model normal variability for the current network) from semantic shift (latent anomalies under augmentation relative to those priors). We will update the figure caption to clarify this intent and emphasize that the t‑SNE panel is illustrative rather than evidentiary.
> >
> > ---
> >
> > ## Q3. Training efficiency
> >
> > A3Point introduces a lightweight, class-wise VQ‑VAE module—implemented with sparse convolutions, a code dimension of $D=64$, and a codebook size of $k=32$ (see architecture in Appendix B.1)—to enable SCP and SSR during training. Below we detail the practical implications and costs:
> >
> > - No inference-time overhead. Both SCP and SSR are training-only components. They are disabled at test time, so inference latency and memory footprint are identical to the baseline model. The exported checkpoint and forward path remain unchanged.
> > - Low-cost statistics via EMA. We maintain per-codeword statistics (variance) using an exponential moving average (EMA). This update is $O(1)$ per iteration with negligible extra memory and does not grow with dataset size or batch length.
> > - Frozen prior for SSR localization. The encoder used to localize SSR regions is kept frozen during localization and does not backpropagate. This design bounds the incremental compute to a lightweight forward pass and avoids gradient overhead, keeping utilization stable.
> >
> > To show the cost clearly, we report a representative runtime snapshot under identical hardware and training settings:
> >
> > | Method   | GPU Memory (MB) | Iterations/s (it/s) | Total Time (h) |
> > | -------- | --------------- | ------------------- | -------------- |
> > | Baseline | 8,496           | 2.72                | 24.4           |
> > | PointDR  | 9,624           | 1.97                | 33.7           |
> > | EAS      | 8,932           | 2.05                | 32.4           |
> > | A3Point  | 17,144          | 1.61                | 41.3           |
> >
> > Given the substantial mIoU gains on SemanticSTF (+9.9 / +11.7), we view the additional training cost as a favorable trade-off: we concentrate budget on training to improve adverse-weather robustness while keeping the inference path clean and unchanged. If desired, the training overhead can be further reduced via standard engineering options (e.g., mixed precision for the VQ‑VAE branch, smaller $k/D$, or alternating SSR localization every $N$ steps), without affecting test-time complexity.

---

> > > ### Author Response · Authors · 2025-11-24
> > > **Reply to Reviewer 79Cq - Part 3**
> > >
> > > ## Q4. Why does Oracle show 0.0 IoU for rare classes while A3Point improves them?
> > >
> > > Revisiting the Oracle results, we find that the 0.0 IoU for motorcycle/motorcyclist on [C] stems from reproducibility instability caused by extreme class imbalance and cross-weather variation. Across multiple reruns, these rare classes yield non-zero IoUs with substantial variance. Below are five independent runs:
> > >
> > > | Run  |  car | bicycle | motorcycle | truck | other-vehicle | person | bicyclist | motorcyclist | road | parking | sidewalk | other-ground | building | fence | vegetation | trunk | terrain | pole | traffic-sign | mIoU |
> > > | ---- | ---: | ------: | ---------: | ----: | ------------: | -----: | --------: | -----------: | ---: | ------: | -------: | -----------: | -------: | ----: | ---------: | ----: | ------: | ---: | -----------: | ---: |
> > > | 1    | 89.4 |    42.1 |        0.0 |  59.9 |          61.2 |   69.6 |      39.0 |          0.0 | 82.2 |    21.5 |     58.2 |         45.6 |     86.1 |  63.6 |       80.2 |  52.0 |    77.6 | 50.1 |         61.7 | 54.7 |
> > > | 2    | 89.6 |    37.4 |        9.6 |  56.4 |          64.3 |   67.3 |      44.8 |          0.8 | 81.8 |    27.9 |     54.6 |         50.1 |     86.5 |  66.2 |       78.3 |  56.3 |    75.4 | 45.8 |         58.6 | 55.3 |
> > > | 3    | 89.7 |    41.8 |        4.0 |  58.6 |          62.8 |   66.1 |      40.7 |          0.0 | 82.0 |    23.4 |     51.8 |         47.2 |     86.7 |  64.9 |       78.7 |  59.1 |    75.0 | 49.0 |         57.0 | 54.6 |
> > > | 4    | 89.6 |    35.3 |        0.2 |  60.8 |          64.3 |   67.4 |      44.4 |          5.4 | 81.3 |    27.4 |     54.6 |         50.2 |     87.2 |  66.3 |       77.8 |  61.1 |    74.0 | 51.8 |         59.5 | 55.7 |
> > > | 5    | 89.7 |    39.9 |        6.8 |  58.1 |          62.1 |   61.6 |      48.5 |         11.1 | 76.4 |    21.7 |     51.4 |         43.5 |     83.1 |  61.3 |       77.2 |  63.4 |    74.9 | 55.3 |         56.4 | 54.9 |
> > >
> > > This instability implies that 0.0 IoU is not systematic failure but a variance artifact under heavy class imbalance. Furthermore, not only A3Point—other methods (including the baseline) also show noticeable gains on motorcycle/motorcyclist. The key driver is data coverage: training on the source domain [A] provides stable and well-represented examples for these categories, which mitigates the scarcity on [C]. As a result, the improvement is not exclusive to any single method; it is a dataset-induced, broadly observable phenomenon.
> > >
> > > Beyond the dataset effect, A3Point adds two mechanisms that are especially beneficial for long-tailed, small-footprint classes under strong perturbations:
> > >
> > > - Exposure via EAS. The augmentation space (EAS) increases diversity in point density and pose, which improves recall for fragile classes that otherwise suffer from sparsity and occlusion at long range.
> > > - Safer supervision via SSR. When augmentations induce semantic shift, SSR switches from hard supervision to latent-space distillation toward the best-matching code. This soft, prior-consistent guidance reduces misalignment between labels and perturbed observations, stabilizing training for rare categories.
> > >
> > > In practice, augmentation-heavy baselines already improve these classes relative to a plain Oracle; A3Point’s SSR/latent guidance further reduces false positives and harmful updates, leading to steadier gains on motorcycles/motorcyclists and similar rare classes. We clarify this distinction—dataset-induced improvements vs. A3Point-specific stabilization—in the revised text.

---

> > > > ### Author Response · Authors · 2025-11-24
> > > > **Reply to Reviewer 79Cq - Part 4**
> > > >
> > > > ## Q5. On the “architecture-agnostic” claim
> > > >
> > > > On the “architecture-agnostic” claim, we adopt more careful wording—“architecture-friendly/broadly applicable”—and provide further clarification:
> > > >
> > > > - Clarifying what we mean by “architecture-agnostic.”:
> > > >   Our intention is not to claim identical performance across paradigms, but to emphasize that our method requires no modifications to the semantic segmentation backbone. It can be plugged into any segmentation network as a training-time module, with zero overhead at inference. To substantiate this, Table 3 reports consistent gains across multiple, diverse backbones without altering their architectures, demonstrating strong architecture compatibility and ease of adoption.
> > > > - Why we did not include projection-/point-based approaches in this paper:
> > > >    The street-scene LiDAR datasets we use exhibit specific characteristics in point density distribution, sensor placement, and coverage. Under these conditions, range-view (projection) and pure point-based paradigms face inherent challenges in modeling long-range sparsity and occlusions. In addition, nearly all prior work on these benchmarks uses voxel-based backbones as the primary comparison axis. To ensure fairness, comparability, and reproducibility, we conducted a systematic evaluation within the community’s mainstream voxel paradigm rather than mixing paradigms. As future work, we plan to extend our experiments to datasets that are better aligned with projection- and point-based methods to further assess cross-paradigm generality.
> > > >
> > > >
> > > > ---
> > > >
> > > > ## Q6. Effectiveness for safely allowing a larger augmentation space
> > > >
> > > > We add experiments to directly evaluate whether A3Point enables “safe expansion” of the augmentation space.
> > > >
> > > > - Scope of the default EAS
> > > >    Our default ranges already span common street-scene LiDAR degradations. Pushing beyond these bounds (e.g., extreme point-drop ratios or jitter) quickly becomes physically implausible and risks label–geometry mismatch, which can corrupt supervision.
> > > >
> > > > - Stress test with arbitrarily large ranges
> > > >    We expand jitter std to [0, 0.10] and point-drop ratio to [0, 0.99] (none–excessive), with all other augmentations unchanged, and evaluate on [A]→[C].
> > > >
> > > >   | Method   | None | light | moderate | heavy | random (light–heavy) | excessive | random* (none–excessive) |
> > > >   | -------- | ---: | ----: | -------: | ----: | -------------------: | --------- | -----------------------: |
> > > >   | Baseline | 31.4 |  37.6 |     38.0 |  37.1 |                 38.7 | 5.1       |                     36.5 |
> > > >   | A3Point  | 31.8 |  38.5 |     40.4 |  40.5 |                 41.3 | 7.8       |                     41.2 |
> > > >
> > > >   As shown in Table 18, we have key observations:
> > > >
> > > >   - Excessive augmentation without SSR is harmful. With very large ranges, the baseline suffers severe degradation, consistent with mis-supervision induced by semantic shift.
> > > >   - SSR stabilizes training under heavy distortions. A3Point maintains strong performance  when sampling randomly across none→excessive, because SSR localizes shifted regions and switches to latent-space distillation to avoid harmful updates.
> > > >   - There is a hard upper bound set by physics. When drop ratio approaches 1 and jitter std approaches 0.1, inputs cease to resemble plausible LiDAR scans; performance drops for all methods, indicating the practical limit of “useful” augmentation.
> > > >
> > > > - Evidence of adaptive behavior
> > > >    As shown in Figure 5, the SSR mask ratio increases with augmentation strength, indicating that the detector adaptively flags more regions as shift intensifies. This adaptivity enables safe use of stronger augmentations: regions consistent with the learned confusion prior receive standard supervision, while shifted regions receive soft, prior-consistent guidance.
> > > >
> > > > In summary, A3Point expands the usable augmentation envelope by mitigating mis-supervision where semantic shift occurs, while respecting a physically grounded ceiling beyond which augmentation is no longer beneficial.

---

> ### Author Response · Authors · 2025-11-24
> **Reply to Reviewer 79Cq - Part 5**
>
> ## Q7. Extension: iterative A3Point with varying augmentation space
>
> We agree that iteratively refining a confusion prior (e.g., a confusion matrix or SCP) is a valuable direction that can improve robustness and generalization.
>
> However, the core limitation of purely offline priors is their static nature: like the multi-model ensemble in Q1, they cannot track the evolving decision boundary, calibration, and capacity of the current network. Consequently, our experiments show that such static priors often misalign with the model’s confidence distribution and error modes—especially early in training—leading to inferior results compared with online evolution.
>
> By contrast, our method updates the confusion prior online on the original domain and freezes it only for anomaly detection on augmented inputs. This design does not require a perfect prior; rather, it maintains alignment with the model’s current state throughout training. Empirically, the adaptive online prior yields better stability and higher final mIoU than offline confusion matrices (see Table 6), with particularly larger gains for long-tailed classes and long-range sparse regions.
>
> To evaluate the reviewer’s suggestion, we further implement an iterative, curriculum-style variant of A3Point that expands the augmentation space in stages. Specifically, we split the training pipeline into three phases: first, we begin with a light augmentation ceiling and train for several epochs; next, we use the resulting SCP as initialization and raise the ceiling to moderate; finally, we proceed to a heavy ceiling in the last phase. In short, the augmentation ceilings progress light → moderate → heavy, and each stage is initialized with the SCP learned in the previous stage. The results are:
>
> | Strategy              | [A] → [C] | [B] → [C] |
> | --------------------- | --------: | --------: |
> | None                  |      38.7 |      24.9 |
> | A3Point               |      41.3 |      27.2 |
> | A3Point+ (curriculum) |      41.5 |      27.8 |
>
> Overall, this iterative scheme adds about +0.2 to +0.6 mIoU over single-stage A3Point when expanding to a larger augmentation space. Therefore, we view it as a promising direction—using SCP statistics to adapt augmentation distributions and SSR thresholds in a closed-loop schedule—while, nevertheless, noting that the principal gains still come from maintaining online alignment rather than from offline aggregation.
>
>
> ---
>
> ## Q8. Clarification of class mapping across datasets
>
> We clarify the exact class mapping in the appendix W. We follow the standard 19-class protocol used by SemanticKITTI/SynLiDAR and SemanticSTF, applying the official mappings to ensure consistency across [A]/[B] (source) and [C] (target).
>
>
> ---
>
> Thank you again for your insightful feedback. We hope our responses and revisions address your concerns. Please do not hesitate to let us know if you have further questions :)

---

### Official Review · Reviewer_wTLP · 2025-11-03

**Soundness:** 2
**Presentation:** 3
**Contribution:** 2
**Rating:** 2
**Confidence:** 4

**Summary:**

This paper focuses on augmentation-based methods among existing approaches for domain generalization under adverse weather conditions. This paper assert that previous augmentation-based methods suffered from the issue that strong augmentations caused semantic shifts, which interfered with training. In this study, the authors propose a framework using VQ-VAE to distinguish between semantic confusion and semantic shift, and they claim to address the problem by introducing a distillation-based loss that prevents semantic shift from disrupting the training dynamics.

**Strengths:**

1.	New idea
2.	Clearly identifies the weaknesses of existing weather-augmentation approaches
3.	The method that separates SSR from SCR is highly sophisticated
4.	Through performance gains and ablations, the authors show strong effectiveness for LiDAR semantic segmentation under adverse weather. Notably, there are large improvements on SynLiDAR → SemanticSTF, where prior methods achieved limited gains.

**Weaknesses:**

1.	The method appears applicable to domain generalization beyond adverse weather; there is no compelling reason it must be evaluated specifically on SemanticSTF.
2.	By the same logic, even if strong augmentation produces SSR, wouldn’t training those regions with the original ground truth still be effective? Why deliberately block cross-entropy learning there? Is there any convincing reason that training with GT in SSR is not good? In real weather conditions, distortions can be just as strong.
3.	It is unclear whether the method actually performs well within SSR regions (i.e., the detected regions during training, or the highly distorted areas within a single sample at inference).
4.	Training time appears to be very long.
5.	Couldn’t a simple prototype-based method separate SCR and SSR without a VQ-VAE? If not, why not? If feasible, that alternative would likely incur much lower training cost.
6.	On LiDARWeather the training schedule uses 15 epochs, but you train for 50. Why? The gains might simply come from longer training.
7.	In Table 1, among the “things” classes, person and motorcyclist presumably have more samples and should be easier to improve, yet they are not; in my knowledge, this is unexpected. Please provide the confusion matrix.

(Minor)
1.	In “Weather-level Comparison” of experiment, please indicate which table the reader should consult.
2.	In Table 1, the experimental results for LiDARWeather and NTN appear to be swapped.
3.	Xiao et al., 2023 primarily proposed SemanticSTF; it is not the work that established point dropping and jittering as the main disturbances in LiDAR data.

**Questions:**

See the above.

---

> ### Author Response · Authors · 2025-11-24
> **Reply to Reviewer wTLP - Part 1**
>
> We sincerely thank you for the thoughtful and constructive review, and for recognizing our contributions in **proposing a new idea**, **clearly identifying the weaknesses of existing augmentation approaches**, **designing a sophisticated method to separate SCR and SSR**, and **demonstrating strong empirical gains**. Below, we address each concern in detail.
>
> ------
> ## Q1: Why evaluate specifically on SemanticSTF?
>
> We agree that our method is conceptually applicable to broader DG scenarios beyond adverse weather. We focus on SemanticSTF for three practical reasons:
>
> - Realism and difficulty: SemanticSTF is one of the few real, densely annotated benchmarks spanning diverse adverse conditions (snow, dense fog, light fog, rain), closely reflecting deployment challenges in autonomous driving.
> - Comparability: Recent methods explicitly targeting weather-robust DG (e.g., LiDARWeather, NTN) report results on SemanticSTF, enabling fair and direct comparisons under a shared protocol.
> - Distortion coverage and stress-testing: Our core contribution is “wide-range augmentation + semantic-shift detection.” In adverse weather, LiDAR degradations can be broadly simulated by point dropping and jittering, and SemanticSTF exhibits strong, spatially non-uniform distortions that directly stress-test our SSR/SCR disentanglement.
>
> To complement SemanticSTF with a controlled, simulation-based setting, we also evaluate on SemanticKITTI-C, which captures synthetic corruption scenarios and corroborates the robustness trends.
>
> ------
>
> ## Q2: Why is training with GT in SSR not desirable?
>
> By construction, SSR denotes regions where strong augmentations induce semantic shift—geometry and density are distorted such that the original labels no longer align with the observed evidence. Supervising these regions with the original GT via cross-entropy therefore injects systematic label noise. We support this with three complementary arguments:
>
> - Empirical evidence: In Table 5, when increasing augmentation from moderate to heavy, the baseline on [A]→[C] drops from 38.0% to 37.1% mIoU. While augmentations generally aid DG, over-aggressive ones hurt—consistent with label-target mismatch in SSR. In contrast, A3Point prevents this drop by masking CE in SSR and applying latent distillation, reaching 40.5% (heavy) and 41.3% (random), thereby mitigating GT noise in SSR.
> - Mechanistic rationale: In SSR, observations deviate from the source-domain semantic manifold (Fig. 2(b)); the original GT is no longer conditionally correct for the augmented evidence. Forcing CE on such regions pulls features toward targets that contradict the observed structure, encouraging spurious fits or collapse. Our latent-space distillation instead uses the global nearest semantic-confusion prior as a soft target, providing a semantically plausible anchor without enforcing potentially incorrect labels, and thus preserving consistency.
> - Ablation support: Table 4 (EAS+SCR vs. EAS+SCR+SSR) shows that merely masking CE in SSR (EAS+SCR) already outperforms EAS alone ([A]→[C]: 38.7→40.2; [B]→[C]: 24.9→26.5), indicating that GT in SSR behaves as systematic noise. Adding SSR distillation yields further gains ([A]→[C]: 40.2→41.3; [B]→[C]: 26.5→27.2), demonstrating that latent-prior distillation is superior to training with GT in SSR.

---

> > ### Author Response · Authors · 2025-11-24
> > **Reply to Reviewer wTLP - Part 2**
> >
> > ## Q3: Does the method actually improve performance within SSR regions?
> >
> > Appendix H provides qualitative evidence that SSR masks substantially overlap with error-prone areas, suggesting that targeting SSR should translate into local improvements. To quantify this more directly, we add two complementary evaluations.
> >
> > **Training-time SSR-region evaluation (approximate mIoU)**:
> >
> > - Setup: We construct a fixed evaluation mask $\mathcal{M}_{SSR}$_eval  on training samples using the same SSR localization rules. To avoid circularity, we fix the prior encoder and variance statistics obtained from the EAS-only model.
> > - Rationale for supervision: Because GT in SSR is unreliable by definition (labels become misaligned after strong augmentation), we follow a semi-supervised protocol and maintain an EMA (mean-teacher) model as a proxy evaluator. The EMA provides a stable, denoised target that tracks the model’s long-term belief without inheriting the instantaneous misalignment introduced by aggressive augmentations. Therefore, higher agreement with the EMA within $\mathcal{M}_{SSR}$_eval indicates improved consistency precisely where GT is noisy.
> > - Result: As shown in Fig. 8 (Appendix J), compared to EAS-only, A3Point’s predictions within $\mathcal{M}_{SSR}$_eval  exhibit higher agreement with the EMA teacher. In particular, adding SSR distillation further increases agreement within SSR regions relative to masking alone.
> >
> > **Inference-time high-distortion subregions evaluation**:
> >
> > - Approximation of high-distortion areas: As shown in Fig. 9 (Appendix J), we approximate “high-distortion” subregions using local density and curvature thresholds:
> >
> >   - or each point $i$, find $k$-NN $N_k(i)$with $k=32$.
> >   - Local $density(i) \approx$inverse of the distance to the $k$-th nearest neighbor.
> >   - Local curvature via PCA on $N_k(i)$: compute the covariance matrix $\sum_i$, obtain eigenvalues $\lambda_1\ge \lambda_2\ge \lambda_3 \ge 0$, define $curvature(i) = \lambda_3 /(\lambda_1+ \lambda_2+ \lambda_3) $ .
> >   - Aggregate point-wise density/curvature to voxels v via mean to obtain  $density(v) $ and  $curvature(v) $.
> >   - Define the high-distortion indicator using quantile-based thresholds (selecting 20% hardest regions): $HighDist(i) = 1(density(i) ≤ τ_d \> || \>  curvature(i) ≥ τ_c)$.
> >
> > - Result: Within these high-distortion masks, A3Point outperforms EAS across the top-5 frequent classes and in overall mIoU, as shown in Table 11:
> >
> >   | Method  | Car  | Road | Sidewalk | Vegetation | Terrain | Overall mIoU |
> >   | ------- | ---- | ---- | -------- | ---------- | ------- | ------------ |
> >   | EAS     | 61.6 | 50.7 | 21.4     | 47.5       | 35.7    | 27.1         |
> >   | A3Point | 76.7 | 57.6 | 26.8     | 51.5       | 37.6    | 34.7         |
> >
> > These analyses indicate that A3Point not only improves global metrics but also delivers meaningful gains within the high-risk regions identified during training, supporting the intended efficacy of SSR localization and supervision.

---

> > > ### Author Response · Authors · 2025-11-24
> > > **Reply to Reviewer wTLP - Part 3**
> > >
> > > ## Q4: Training-time overhead
> > >
> > > A3Point introduces a lightweight, class-wise VQ‑VAE module—implemented with sparse convolutions, a code dimension of $D=64$, and a codebook size of $k=32$ (see architecture in Appendix B.1)—to enable SCP and SSR during training. Below we detail the practical implications and costs:
> > >
> > > - No inference-time overhead. Both SCP and SSR are training-only components. They are disabled at test time, so inference latency and memory footprint are identical to the baseline model. The exported checkpoint and forward path remain unchanged.
> > > - Low-cost statistics via EMA. We maintain per-codeword statistics (variance) using an exponential moving average (EMA). This update is $O(1)$ per iteration with negligible extra memory and does not grow with dataset size or batch length.
> > > - Frozen prior for SSR localization. The encoder used to localize SSR regions is kept frozen during localization and does not backpropagate. This design bounds the incremental compute to a lightweight forward pass and avoids gradient overhead, keeping utilization stable.
> > >
> > > To show the cost clearly, we report a representative runtime snapshot under identical hardware and training settings:
> > >
> > > | Method   | GPU Memory (MB) | Iterations/s (it/s) | Total Time (h) |
> > > | -------- | --------------- | ------------------- | -------------- |
> > > | Baseline | 8,496           | 2.72                | 24.4           |
> > > | PointDR  | 9,624           | 1.97                | 33.7           |
> > > | EAS      | 8,932           | 2.05                | 32.4           |
> > > | A3Point  | 17,144          | 1.61                | 41.3           |
> > >
> > > Given the substantial mIoU gains on SemanticSTF (+9.9 / +11.7), we view the additional training cost as a favorable trade-off: we concentrate budget on training to improve adverse-weather robustness while keeping the inference path clean and unchanged. If desired, the training overhead can be further reduced via standard engineering options (e.g., mixed precision for the VQ‑VAE branch, smaller $k/D$, or alternating SSR localization every $N$ steps), without affecting test-time complexity.
> > >
> > > ------
> > >
> > > ## Q5: Could a simple prototype-based method replace VQ-VAE?
> > >
> > > We implement a class-wise K-prototype variant ($k = 32$) as a drop-in alternative. Under the same training budget, VQ-VAE consistently performs better, as shown in Table 13:
> > >
> > > | Setting             | [A]→[C] | [B]→[C] |
> > > | ------------------- | ------- | ------- |
> > > | None                | 31.4    | 15.5    |
> > > | EAS                 | 38.7    | 24.9    |
> > > | Prototype-based     | 40.0    | 25.4    |
> > > | VQ-VAE-based (ours) | 41.3    | 27.2    |
> > >
> > > VQ-VAE outperforms the prototype baseline by 1.3–1.8 mIoU. We attribute this gap to three advantages:
> > >
> > > - Intra-class multimodality. Error/ambiguity patterns within a class are inherently multi-modal (e.g., lane markings near curbs, partial occlusions, and far-range sparsity). Per-class codebooks in VQ-VAE capture multiple modes more faithfully than a single prototype centroid, leading to more accurate localization of risky subregions.
> > > - Well-formed anomaly modeling. VQ-VAE enables per-codeword variance tracking (via EMA), which yields compact, approximately Gaussian local clusters and an interpretable “prior radius” for anomaly detection. In contrast, prototype methods often require extra metric-learning losses or carefully tuned thresholds to behave well, and can be brittle across conditions.
> > > - Online adaptivity with stability: Our prior is updated online with the evolving backbone, while SSR localization uses a frozen encoder for backprop, providing statistics that are both responsive and stable. Prototype updates may lag or drift as the feature space shifts, degrading SSR localization over training.
> > >
> > > In short, VQ-VAE offers a more robust, interpretable, and empirically stronger mechanism for modeling class-wise priors, delivering consistent gains at only modest additional training cost.

---

> > > > ### Author Response · Authors · 2025-11-24
> > > > **Reply to Reviewer wTLP - Part 4**
> > > >
> > > > ## Q6: 50 epochs vs. LiDARWeather’s 15 epochs
> > > >
> > > > For consistency and internal fairness, we adopt a PointDR-like 50-epoch schedule for all main results and ablations (Tables 3–6). To disentangle schedule length from algorithmic gains, we conduct controlled comparisons under both short (15-epoch, LiDARWeather-style) and long (50-epoch) schedules.
> > > >
> > > > - Short schedule (15 epochs): With identical training budgets, A3Point still surpasses our LiDARWeather reproduction by +2.7–4.5 mIoU, indicating that the gains are not contingent on extended training.
> > > >
> > > >   | Method         | [A]→[C] | [B]→[C] |
> > > >   | -------------- | ------- | ------- |
> > > >   | Baseline       | 31.4    | 15.5    |
> > > >   | LiDARWeather*  | 37.8    | 20.4    |
> > > >   | A3Point (ours) | 40.5    | 25.9    |
> > > >
> > > > - Long schedule (50 epochs): Extending Baseline and LiDARWeather to 50 epochs yields only modest improvements, and both remain below A3Point, indicating that schedule length alone does not close the gap.
> > > >
> > > >   | Method         | [A]→[C] | [B]→[C] |
> > > >   | -------------- | ------- | ------- |
> > > >   | Baseline*      | 32.3    | 16.1    |
> > > >   | LiDARWeather*  | 38.6    | 21.6    |
> > > >   | A3Point (ours) | 41.3    | 27.2    |
> > > >
> > > > Together, these results demonstrate that A3Point’s advantages arise from its augmentation-aware latent learning and SSR handling, rather than from longer training schedules.
> > > >
> > > >
> > > >
> > > > ------
> > > >
> > > > ## Q7: Confusion matrix for “things” classes
> > > >
> > > > We appreciate the suggestion. To clarify the relationship between class frequency and improvement, we add confusion matrices for both the Oracle and A3Point on SemanticSTF in Appendix N. Our key observations are:
> > > >
> > > > - Frequency is not a sufficient predictor of gains. Although classes such as person and motorcyclist appear relatively often, they frequently suffer from severe occlusions, pose/shape variability, and far-range sparsity—reducing the number of effective, high-quality points per scene. Under adverse weather, these factors exacerbate inter-class confusions (e.g., person vs. pole/sign; motorcyclist vs. bicycle/motorcycle), which limits attainable improvements even when the aggregate frequency is nontrivial.
> > > >
> > > > - Larger gains tend to occur for classes with more stable and distinctive local geometry. Categories such as bicycle, motorcycle, traffic sign, and car preserve clearer geometric signatures under weather perturbations (e.g., edges, planar surfaces, repeatable part configurations). Our SCP/SSR priors help the model maintain consistency in these regions, translating into stronger improvements, as reflected in Tables 1–2.
> > > >
> > > >
> > > >
> > > > ------
> > > >
> > > > ## Q8: Minor issues
> > > >
> > > > Thank you for the helpful feedback. We  implement the following clarifications and edits:
> > > >
> > > > - Clear cross-references: In the “Weather-level Comparison” section, we now explicitly reference Table 1 ([A]→[C]) and Table 2 ([B]→[C]) so readers can quickly locate the relevant results.
> > > > - Consistent ordering of methods: We verify that the citations for LiDARWeather and NTN are correct and standardized the method order by publication date. We keep the same ordering in both Table 1 and Table 2 for consistency—even though NTN underperforms LiDARWeather in Table 1 but achieves the best results in Table 2. This avoids per-table reordering by performance and reduces potential confusion.
> > > > - Citation clarification: In Related Work, we revise the description of Xiao et al. (2023) to more accurately reflect its main contribution—introducing the SemanticSTF benchmark. While the paper also presents an augmentation-based baseline and highlights “geometric perturbation and point dropping” as primary degradation modes, we now position it within a broader and more balanced survey, avoiding over-emphasis on it as a standalone method contribution.
> > > >
> > > > ---
> > > >
> > > > Thank you again for your careful review and helpful suggestions. We hope the proposed clarifications, additional analyses, and ablations in the revision can resolve your concern. Please do not hesitate to let us know if you have further questions :)

---

> > > > > ### Comment · Reviewer_wTLP · 2025-11-27
> > > > > **post rebuttal**
> > > > >
> > > > > The authors’ rebuttal substantially strengthens the submission through new ablations, region-level SSR analyses, prototype comparisons, and detailed training-cost clarification. These additions directly address earlier concerns and provide solid empirical grounding for the method’s design choices. Although some questions on broader generalization remain, the core contribution is now clearer and better supported. Overall, I will update my score accordingly.

---

> ### Author Response · Authors · 2025-11-27
>
> Dear Reviewer wTLP,
>
> Thank you very much for your thoughtful and constructive feedback. We truly appreciate your careful consideration of our rebuttal and are glad to hear that the additional ablations, region-level SSR analyses, prototype comparisons, and training-cost clarifications helped address the concerns raised in the initial review.
>
> We’re sincerely grateful for your updated evaluation. Your insights have been invaluable in strengthening the presentation and empirical justification of our method, and we will make sure to incorporate the key clarifications into the final version of the paper.
>
> Thank you again for your time and support!
>
> Best regards,
>
> Authors

---

### Author Response · Authors · 2025-11-24
**General Response to Reviewers and Revision Submitted**

We would like to thank all of the reviewers for their constructive and valuable feedback on our work,  We provide point-by-point responses to each reviewer's concerns and upload a revised PDF. The changes made to the PDF are summarized as follows:



**Main manuscript**

- We add references to the tables for better clarity  (**[wTLP]**).
- We revise the description in related work to provide a more appropriate characterization (**[wTLP]**).
- We fix  typos in Table 1 caption, Appendix C/G, Appendix I/J  (**[My2N]**).

**Appendix**

- We  summarize all additional experimental results, including:
  - Table 11, Appendix J: High-distortion subregion evaluation (**[wTLP]**).
  - Table 12, Appendix K: Training-time overhead comparison (**[wTLP]**, **[79Cq]**, **[GuJ4]**, **[My2N]**).
  - Table 13, Appendix L: Comparison with a Prototype-based Alternative  (**[wTLP]**).
  - Table 14-15, Appendix M: Ablation with training schedule  (**[wTLP]**).
  - Table 16, Appendix O: Comparison with ensemble confusion priors  (**[79Cq]**).
  - Table 17, Appendix P: Multiple independent runs  (**[79Cq]**).
  - Table 18, Appendix Q: Stress test with enlarged augmentation range (**[79Cq]**).
  - Table 19, Appendix R: Extension of A3Point (**[79Cq]**).
  - Table 20, Appendix S: Stronger augmentation for previous methods (**[GuJ4]**).
  - Table 21, Appendix U: Comparisons with other DG methods (**[My2N]**).
  - Table 22, Appendix V: Controlled comparisons of SSR supervision (**[My2N]**).
- We add additional visualizations, including:
  - Figure 8, Appendix J: Agreement with EMA teacher within $\mathcal{M}_{\mathrm{SSR}\_\mathrm{eval}}$ (**[wTLP]**).
  - Figure 9, Appendix J: Example scenes showing “high-distortion” subregions. (**[wTLP]**).
  - Figure 10, Appendix N: Confusion matrices. (**[wTLP]**).
  - Figure 11, Appendix T: t-SNE of the learned VQ-VAE codebook. .(**[GuJ4]**).
- We provide class mapping across datasets in Appendix W (**[79Cq]**).

We hope that our revised manuscript and responses will have a positive impact on the final evaluation. Once again, we express our gratitude to all reviewers for their efforts in this review process and look forward to further discussions.

---

### Author Response · Authors · 2025-12-02
**Author Final Remarks**

We sincerely thank the AC and all reviewers for their careful evaluation, constructive feedback, and engaging discussions. We appreciate the recognition of our work’s **problem formulation, methodological novelty, and consistent empirical gains across adverse-weather DG settings.**

---
**Key Contributions**

Our work introduces A3Point, an adaptive augmentation-aware latent learning framework for robust LiDAR semantic segmentation under adverse weather. It addresses the long-standing dilemma of leveraging a large augmentation space without incurring semantic shift.
- **Adaptive use of a large augmentation space**：We broaden point-drop and jitter ranges to cover mild→heavy perturbations (and stress-test none→excessive), while safeguarding training via region-adaptive supervision so stronger augmentations remain beneficial rather than harmful.
- **Two-step disentanglement of semantic confusion and semantic shift**：SCP latent learning (per-class VQ-VAE) captures discrete confusion patterns from unaugmented predictions with reconstruction guarantees; SSR localization uses a frozen prior encoder with EMA variances to detect latent anomalies on augmented predictions, separating SCR and SSR; SCR uses CE with GT, SSR uses class-agnostic latent distillation to the global nearest code.
- **Strong and consistent results**：SOTA on SemanticKITTI→SemanticSTF and SynLiDAR→SemanticSTF (+9.9/+11.7 mIoU over Minkowski baselines), consistent gains on SPVCNN and SemanticKITTI-C, with zero inference-time overhead since SCP/SSR are training-only.

---
**Concerns Addressed**

We summarize how we address them in the revision (with new analyses, ablations, visualizations, and clarifications):
- Applicability and benchmarks: We focus on SemanticSTF for realism/comparability and corroborate with SemanticKITTI-C; the framework is broadly applicable to DG.
- Training with GT in SSR: GT becomes noisy under shift; masking helps, and latent-prior distillation performs best (Table 4).
- Effectiveness within SSR/high-distortion regions: We add EMA-teacher agreement in SSR (Fig. 8) and a high-distortion subregion evaluation (Table 11), both favoring A3Point.
- Training cost: We report runtime and memory comparisons (Table 12); the extra cost is training-only and inference is identical; we outline practical cost-reduction options.
- Prototype alternative: A class-wise K-prototype prior underperforms VQ-VAE (Table 13) due to limited multimodality, weaker anomaly modeling, and poorer online adaptivity.
- Schedule fairness: Under 15- and 50-epoch schedules, A3Point consistently outperforms LiDARWeather with identical budgets (Tables 14–15).
- Universality/model dependency of confusion: Offline priors (single/ensemble) vs. our online prior (Table 16); online co-evolution with the backbone yields the strongest results.
- t-SNE interpretation: t-SNE is illustrative only; SSR detection relies on latent anomaly checks against class-wise priors learned on unaugmented data and frozen for augmented inputs.
- Distillation vs. removal in SSR: Latent-prior distillation preserves informative gradients and outperforms simple removal (Table 4).
- Stronger augmentation for prior methods: Equipping baselines with EAS improves them but they saturate below A3Point without explicit shift handling (Table 20).
- Comparisons to other DG families: Mean Teacher and SCE yield modest gains; A3Point remains best via explicit confusion–shift disentanglement (Table 21).
- Global nearest code choice: Global > soft-CE > class-conditional > pre-aug consistency (Table 22), avoiding bias from potentially incorrect labels.
- Rare-class Oracle 0.0 IoU: Multi-run variance explains instability; broader EAS exposure and safer SSR supervision stabilize long-tailed classes (Table 17).
- Safe expansion of augmentation space: Stress tests from none→excessive confirm SSR stabilizes training and reveal a physics-grounded ceiling (Table 18).
- Architecture generality and clarity: We adopt “architecture-friendly/broadly applicable,” show gains on Minkowski and SPVCNN (Table 3), and add class mappings (Appendix W), codebook t-SNE (Fig. 11), things-class confusion matrices (Fig. 10), SSR visualizations, and algorithm pseudocode.

---
**Discussion Summary**

In the earlier discussion phase (i.e., prior to the information leak), two reviewers had already participated. Reviewer **[wTLP]** stated concerns were resolved by our added ablations, SSR analyses, prototype comparisons, and cost clarifications, and **updated the score from 2 to 6**; Reviewer **[GuJ4]** maintained a positive score following our responses; overall, **reviewers show a consistent inclination toward acceptance.**

---
We believe A3Point provides a principled, practical path to safely leveraging strong augmentations by explicitly modeling and handling semantic shift, and we hope it contributes meaningfully to robust LiDAR perception under adverse weather.

Thank you again for your time and consideration during the review.

---

### Meta-Review · Area_Chair_D6qf · 2025-12-26

**Summary:**

The paper focuses on domain generalization for LiDAR semantic segmentation under adverse weather, identifying semantic confusion versus augmentation-induced semantic shift. It designs an adaptive latent framework with region-aware supervision, achieving good gains across benchmarks. The approach is novel, well-motivated, empirically strong.
Based on the feedback from reviewers, the decision was made to recommend it for acceptance. We congratulate the authors on their acceptance!
On the other hand, authors should revise the paper taking into account the reviewers' comments, such as the issues and concerns mentioned in Weaknesses.

**Reviewer Concerns:**

Concerns are training efficiency, alternative simpler designs, and deeper analysis of specific class behaviors. These issues are largely analytical or implementation-related, do not undermine the core idea, and are addressable through clarification or future work.

**Reviewer Scores:**

Reviewers’ scores range from reject to strong accept, with most reviewers rating soundness, presentation, and effectiveness positively. Multiple reviewers consider the work above the acceptance threshold and acknowledge its novelty and impact. Overall consensus supports acceptance despite some reservations.

---

### Decision · Program_Chairs · 2026-01-26

Accept (Poster)